# Host-derived *Lactobacillus plantarum* alleviates hyperuricemia by improving gut microbial community and hydrolase-mediated degradation of purine nucleosides

Yang Fu[1], Xiao-Dan Luo[1], Jin-Ze Li[1], Qian-Yuan Mo[1], Xue Wang[2], Yue Zhao[1], You-Ming Zhang[2], Hao-Tong Luo[1], Dai-Yang Xia[3], Wei-Qing Ma[1], Jian-Ying Chen[1], Li-Hau Wang[1], Qiu-Yi Deng[1], Lukuyu Ben[4], Muhammad Kashif Saleemi[5], Xian-Zhi Jiang[6], Juan Chen[6], Kai Miao[7], Zhen-Ping Lin[8], Peng Zhang[9], Hui Ye[1], Qing-Yun Cao[1], Yong-Wen Zhu[1], Lin Yang[1], Qiang Tu[10,11]*, Wence Wang[1]*

[1]State Key Laboratory of Swine and Poultry Breeding Industry, College of Animal Science, South China Agricultural University, Guangzhou, China; [2]State Key Laboratory of Microbial Technology, Shandong University, Shandong, China; [3]School of Marine Sciences, Sun Yat-sen University, and Southern Marine Science and Engineering Guangdong Laboratory, Zhuhai, China; [4]International Livestock Research Institute, Nairobi, Kenya; [5]Department of Pathology, University of Agriculture Faisalabad, Faisalabad, Pakistan; [6]Microbiome Research Center, Moon (Guangzhou) Biotech Co. Ltd, Guangdong, China; [7]CancerCenter, Faculty of Health Sciences, University of Macau, Macau, China; [8]Shantou Baisha Research Institute of Origin Species of Poultry and Stock, Shantou, China; [9]Chimelong Safari Park, Chimelong Group Co, Guangzhou, China; [10]Helmholtz International Lab for Anti-Infectives, Shandong University–Helmholtz Institute of Biotechnology, State Key Laboratory of Microbial Technology, Shandong University, Qingdao, China; [11]Shenzhen Key Laboratory of Genome Manipulation and Biosynthesis, CAS Key Laboratory of Quantitative Engineering Biology, Shenzhen Institute of Synthetic Biology, Shenzhen Institute of Advanced Technology, Chinese Academy of Sciences, Shenzhen, China

*For correspondence:
qiang.tu@siat.ac.cn (QT);
wangwence@scau.edu.cn (WW)

**Abstract** The gut microbiota is implicated in the pathogenesis of hyperuricemia (HUA) and gout. However, it remains unclear whether probiotics residing in the host gut, such as *Lactobacillus*, can prevent HUA development. Herein, we isolated *Lactobacillus plantarum* SQ001 from the cecum of HUA geese and conducted in vitro assays on uric acid (UA) and nucleoside co-culture. Metabolomics and genome-wide analyses, revealed that this strain may promote nucleoside uptake and hydrolysis through its nucleoside hydrolase gene. The functional role of *iunH* gene was confirmed via heterologous expression and gene knockout studies. Oral administration of *L. plantarum* SQ001 resulted in increased abundance of *Lactobacillus* species and reduced serum UA levels. Furthermore, it downregulated hepatic xanthine oxidase, a key enzyme involved in UA synthesis, as well as renal reabsorption protein GLUT9, while enhancing the expression of renal excretion protein ABCG2. Our findings suggest that *L. plantarum* has potential to ameliorate gut microbial dysbiosis with HUA, thereby offering insights into its potential application as a probiotic therapy for individuals with HUA or gout.

## Editor's evaluation

Multiple studies have shown that the gut microbiota is involved in the metabolism of uric acid and influences systemic uric acid levels. However, the specific bacteria and genes involved in this process are not known. The current study provides convincing evidence that *Lactobacillus* plantarum prevents the synthesis of uric acid through the hydrolysis of nucleosides through the nucleoside hydrolase gene (iunH). This marks a valuable contribution to the field.

## Introduction

Hyperuricemia (HUA) is a metabolic disorder characterized by elevated serum uric acid (UA) levels exceeding 420 μmol/l (7 mg/dl), and affecting 8–25% of the global population (*Li et al., 2022*). The incidence of HUA is on the rise, especially among younger populations, driven by diets high in purine and glucose. Moreover, HUA has been linked to various health risks including gout, renal injury, diabetes, hypertension, and cardiovascular disease, thereby posing a significant public health concern (*Johnson et al., 2018*; *Zhang et al., 2019a*).

The regulation of HUA primarily involves three organs: the liver, kidney, and the intestine (*Dalbeth et al., 2021*). The upregulation of hepatic purine metabolism and inhibition of renal UA excretion play pivotal roles in the pathogenesis of HUA, while the intestine contributes to both UA production and excretion (*Dalbeth et al., 2021*; *Niu et al., 2018*). The gut accounts for approximately one-third of total urate excretion. Upon secretion into the gut, UA undergoes metabolism by gut microbes. Therefore, dysbiosis in the gut microbiota is closely associated with dysregulation of urate degradation and systematic inflammation in the host (*Yun et al., 2017*). However, it remains unclear how specific gut microbes influence elevated levels of UA and inflammation in the host.

Probiotics and prebiotics have emerged as promising natural therapies due to their beneficial effects on gut microbiota regulation and overall health enhancement (*Zhao et al., 2022b*; *Wang et al., 2022*). Modulating the gut microbiota and maintaining intestinal balance through probiotics, prebiotics, and even fecal transplants may represent a novel approach to managing HUA from a micro-ecological perspective. *Lactobacillus* is a well-known probiotic that has been reported to utilized for modulating the gut microbiota and alleviating HUA (*Cao et al., 2022a*). Previous studies have demonstrated two mechanisms involved in the alleviation of HUA by *Lactobacillus*. The first one is directly degrading UA (*Wu et al., 2021*), and the other is degrading nucleosides, the precursors of UA in the gut (*Cao et al., 2022b*; *Li et al., 2014*; *Hsu et al., 2019*). However, the specific mechanisms by which these *Lactobacillus* spp. degrade UA or nucleosides remain incompletely understood. Recent investigations have indicated that ribonucleoside hydrolase RihA-C and the nucleoside hydrolase Nhy69 from *Lactobacillus*, which cloned and recombinant, shed light on a potential degradation mechanism within *Lactobacillus* (*Li et al., 2023a*; *Li et al., 2023b*). Nevertheless, there is limited evidence regarding gene knockout validation in *Lactobacillus* and the alleviation of host-derived *Lactobacillus* in HUA is still unknown.

The establishment of reliable and stable animal models for HUA has been a subject of debate among researchers. Mouse models are extensively utilized in HUA or gout studies, but they pose certain challenges as previous studies described. In addition to variances in uricase activity and urate concentration, the mouse modeling form also varies from the development of HUA in humans (*Lu et al., 2019*). Prior research has demonstrated that geese serve as an excellent model for investigating HUA and gout (*Wang et al., 2021*; *Fu et al., 2024*; *Xi et al., 2019*; *Dalbeth et al., 2016*).

Here, we established an HCP (High calcium and protein) diet-induced HUA geese model, and isolated a *Lactobacillus plantarum* strain, SQ001, from HUA geese. Its nucleoside degradation function was investigated through in vitro experiments compared to commercial strains. Genome-wide analysis revealed four genes (*iunH*, *yxjA*, *rihA*, and *rihC*) in *L. plantarum* SQ001 that potentially contribute to nucleoside degradation. We subsequently confirmed the nucleosides hydrolase function of *iunH* through heterologous expression in *E. coli* and gene knockout in *L. plantarum* SQ001. Oral gavage of *L. plantarum* SQ001 further validated the alleviating effect of *L. plantarum* SQ001 in the HUA geese and mouse model. In conclusion, our findings demonstrated that host-derived gut microbes could alleviate gut microbial dysbiosis and HUA in hosts, and provided insights into the potential therapeutic use of *L. plantarum* for improving HUA or gout.

**eLife digest** Our blood contains many components, including waste products that need to be transported to the kidneys, where they can exit the body through urine. One such molecule, known as uric acid, forms when cells break down old DNA and other similar molecules. This process has several steps, with DNA being broken down into intermediate molecules called nucleosides before being converted into uric acid.

If the amount of uric acid in the bloodstream becomes too high (a condition known as hyperuricemia), humans and other animals can develop high blood pressure, chronic kidney disease and other illnesses. Recent studies suggest that some types of 'friendly' bacteria living in the human gut may influence how uric acid levels are regulated. But the precise role these bacteria play remains unclear.

Here, Fu, Luo et al. isolated one of these friendly bacterial species, known as *Lactobacillus plantarum*, from the gut of geese with hyperuricemia. The team grew the bacteria in the laboratory in two environments: one containing uric acid and the other containing nucleosides. The experiments revealed that while *Lactobacillus plantarum* does not directly act on uric acid, it does have enzymes that can convert nucleosides into other molecules.

Further investigations, using whole-genome and metabolomic analyses, showed that *Lactobacillus plantarum* contains three genes encoding enzymes that act on a type of nucleoside known as a purine. Knocking out one of these genes prevented the bacteria from being able to convert purines into other molecules. Subsequently, Fu, Luo et al. demonstrated that *Lactobacillus plantarum* helps to mitigate the effects of hyperuricemia in geese and mice.

These findings provide valuable insights into how microbes living in the gut regulate uric acid levels in their hosts. They may also inform future strategies for preventing and treating hyperuricemia in humans.

## Results

### Gut microbiota differences in diet-dependent HUA

To investigate the potential effects of gut microbiota, geese were assigned to either a normal or HCP diet and maintained on their respective diets for 28 consecutive days (*Figure 1a*), food recipes and nutrient levels of experimental diets refer to our previous article (*Fu et al., 2024*). In geese fed the HCP diet for 28 days, markedly worsened growth performance was shown by lower final average weight (p < 0.0001), higher mortality (p < 0.0001), and higher relative weight of liver (p = 0.0254; *Figure 1b–d*). Moreover, the serum biochemical results indicated the successful construction of the HUA model. Serum UA (p < 0.0001), xanthine oxidase (XOD; p < 0.0001), blood urea nitrogen (BUN; p < 0.0001), and creatinine (p < 0.0001) were significantly elevated in the HUA group, with UA exceeding the threshold (420 µmol $l^{-1}$; *Figure 1e, f*).

To investigate the impact of dietary patterns on gut microbiome remodeling, 16S ribosomal RNA (rRNA) gene sequencing of cecal chyme from CON and HUA geese was conducted after 28 days. The HCP diet caused a slight but significant decrease in the alpha diversity of gut microbiota in geese (Chao index, p = 0.0027; Shannon index, p = 0.0100; *Figure 2a, b*). Analysis of beta diversity using the Bray–Curtis dissimilarity metric in principal coordinate analysis (PCoA) showed a clear segregation of gut microbial signatures between CON and HUA groups (p = 0.0040; *Figure 2c*). LDA (linear discriminant analysis) effect size (LEfSe) analysis of the taxonomic changes revealed that the HCP diet led to distinct bacterial enrichment. The main bacterial families present in geese on a normal diet were *Butyricicoccaceae* (p = 0.0003), *Ruminococcaceae* (p = 0.0103), and *Lactobacillaceae* (p = 0.0003). Conversely, the species *Ruminococcus torques* group (p < 0.0001) and *Ruminococcus gauvreauil* group (p = 0.0005) showed significant proliferation when geese were fed an HCP diet (*Figure 2d, e* and *Figure 2—figure supplement 1A–C*).

These findings collectively indicate a significant impact of the HCP diet on gut microbial remodeling, particularly reflected in the changes in *Lactobacillus* and *Butyricicoccus* abundances. Furthermore, based on Phylogenetic Investigation of Communities by Reconstruction of Unobserved States (PICRUSt) prediction analysis, the abundance profile of the Kyoto encyclopedia of genes and genomes (KEGG) pathway revealed that purine metabolism (p = 0.0002) and Mitogen-activated protein kinase

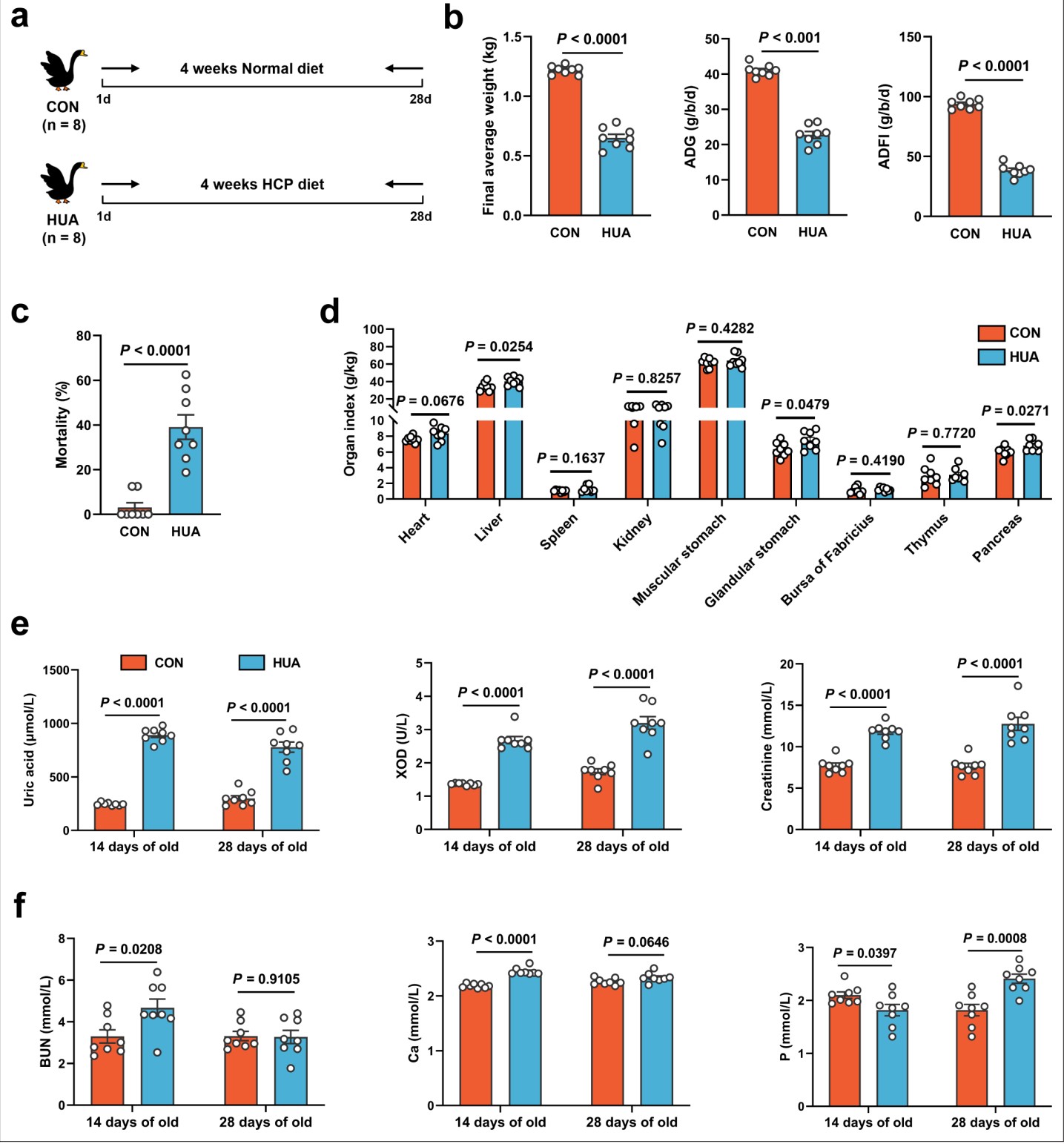

**Figure 1.** Growth and metabolism differences in diet-dependent hyperuricemia (HUA). (**a**) Experimental design. (**b**) Final average weight, average daily feed intake (ADFI), and average daily gain (ADG) of geese after 28 days of control (CON, *n* = 8) or HCP diet (HUA, *n* = 8). (**c**) Mortality of geese after 28 days of control (CON, *n* = 8) or HCP diet (HUA, *n* = 8). (**d**) The organ index of geese after 28 days of control (CON, *n* = 8) or HCP diet (HUA, *n* = 8). (**e**) The serum uric acid (UA), xanthine oxidase (XOD), and creatinine levels of geese after 14 or 28 days of control (CON, *n* = 8) or HCP diet (HUA, *n* = 8). (**f**) The blood urea nitrogen (BUN), calcium (Ca), and phosphorus (P) levels of geese after 14 or 28 days of control (CON, *n* = 8) or HCP diet (HUA, *n* = 8). Statistical significance in (**b–f**) was determined by unpaired two-tailed Student's *t*-test. Data with error bars represent mean ± s.e.m.

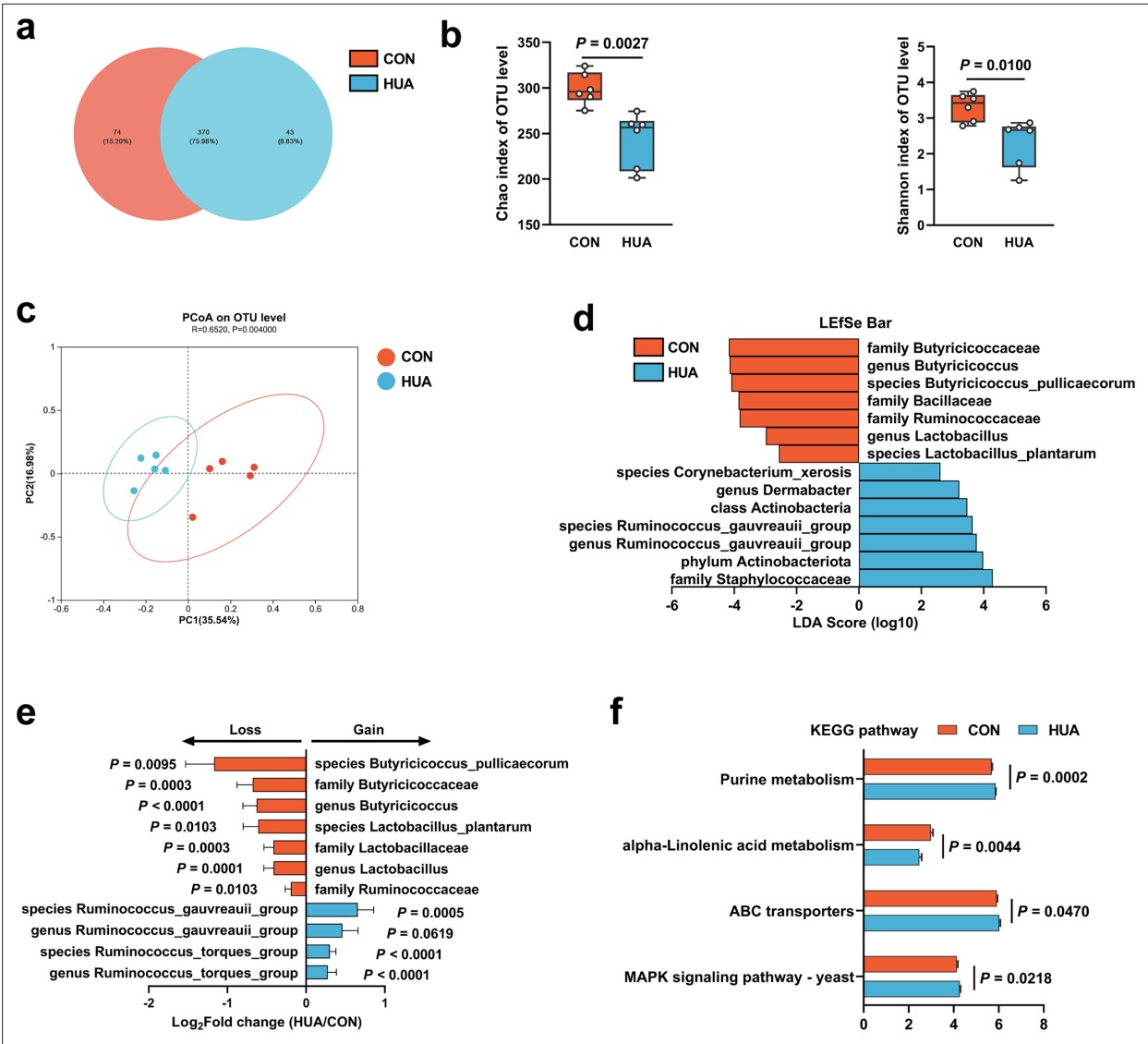

**Figure 2.** Gut microbiota differences in diet-dependent hyperuricemia (HUA). (**a**) Venn plot of bacterial operational taxonomic unit (OTU) in fecal samples of geese fed with control diet (CON, 15.20%) or HCP diet (HUA, 8.83%) after 28 days. (**b**) Chao and Shannon indices of indicated groups based on alpha diversity analysis (*n* = 6). (**c**) Principal component analysis of bacteria on OTU level with 95% confidence regions between CON and HUA groups (*n* = 5, *r* = 0.6520, p = 0.0040). (**d**) Linear discriminative analysis (LDA) score obtained from LDA effect size (LEfSe) analysis of fecal microbiota in CON and HUA groups. Bacterial with Kruskal–Wallis ≤0.05, as well as LDA >2, are reported. (**e**) The alteration trends of the bacterial relative abundance (*n* = 5). The x-axis shows the log2 fold change of the bacterial relative abundance in the HUA group compared to the CON group. (**f**) Summary of major pathway changes based on Phylogenetic Investigation of Communities by Reconstruction of Unobserved States (PICRUSt) prediction analysis of 16S data from fecal samples of CON and HUA groups. Pathways with p ≤ 0.05 by Wilcoxon rank-sum test are reported. Statistical significance in (**b**) was determined by unpaired two-tailed Student's *t*-test. Data with error bars represent mean ± s.e.m.

The online version of this article includes the following figure supplement(s) for figure 2:

**Figure supplement 1.** Relative abundance of gut microbes in diet-dependent hyperuricemia (HUA).

(MAPK) signaling pathway (p = 0.0218) in gut microbiota has increased significantly, while alpha-linolenic acid metabolism has decreased considerably (*Figure 2f*). The overall findings of this study suggested that HUA in geese has a detrimental impact and leads to alterations in the composition of gut microbiota.

## *L. plantarum* SQ001 hydrolyze nucleosides in vitro

We acquired a strain, *L. plantarum* SQ001, from a cecal chyme sample collected from HUA geese. Whole-genome sequencing of *L. plantarum* SQ001 was performed and a completed genome was

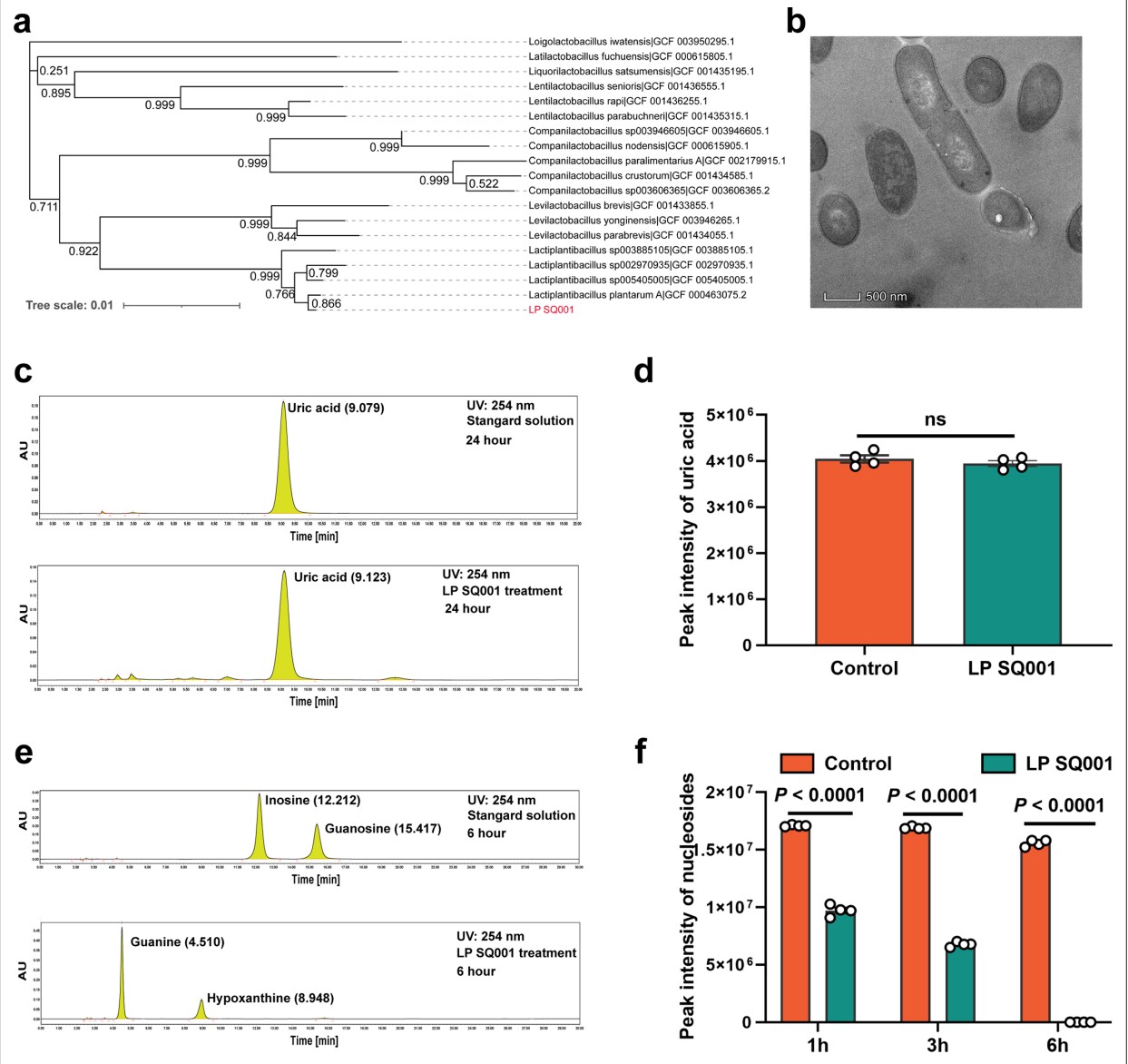

**Figure 3.** *L. plantarum* SQ001 can hydrolyze nucleosides in vitro. (**a**) The phylogenetic tree of *L. plantarum* SQ001. 16S ribosomal RNA (rRNA) of *L. plantarum* SQ001 was extracted to sequence and blast the phylogeny, compared with 18 strains from different origins. Bootstrap values are displayed under the branches. (**b**) The transmission electron microscope results of the strain *L. plantarum* SQ001. Magnification of 12,000. (**c**) Degradation of uric acid by *L. plantarum* SQ001 from high-performance liquid chromatography (HPLC) analysis. (**d**) Peak intensity of uric acid by *L. plantarum* SQ001 after 24 hr treatment (*n* = 4). (**e**) Degradation of inosine and guanosine by *L. plantarum* SQ001 from HPLC analysis. (**f**) Peak intensity of nucleosides by *L. plantarum* SQ001 after 1, 3, and 6 hr treatment (inosine and guanosine, *n* = 4). Statistical significance in (**e**) was determined by unpaired two-tailed Student's *t*-test. Data with error bars represent mean ± s.e.m. LP SQ001, *L. plantarum* SQ001; LP 23180, *L. plantarum* 23180.

obtained. Comparing it with the 18 strains from the RefSeq, *L. plantarum* SQ001 was closer to *L. plantarum* A (GCF_000463075.2, 0.866) (*Figure 3a*). Scanning electron microscopy showed it to be long rod-shaped (*Figure 3b*). To elucidating the mechanism by which *L. plantarum* SQ001-mediated alleviation in HUA, the ability of *L. plantarum* SQ001 to metabolize UA was initially investigated. *L. plantarum* SQ001 has no effect on the degradation of UA (*Figure 3c, d*), but it can metabolize purine nucleosides (inosine and guanosine), the precursors of UA production. Results indicated that *L. plantarum* SQ001 completely absorbed or hydrolyzed both purine nucleosides within 6 hr (p < 0.0001; *Figure 3e, f*). These findings indicated that *L. plantarum* SQ001 alleviates HUA not through direct degradation of UA, but by metabolizing its precursors.

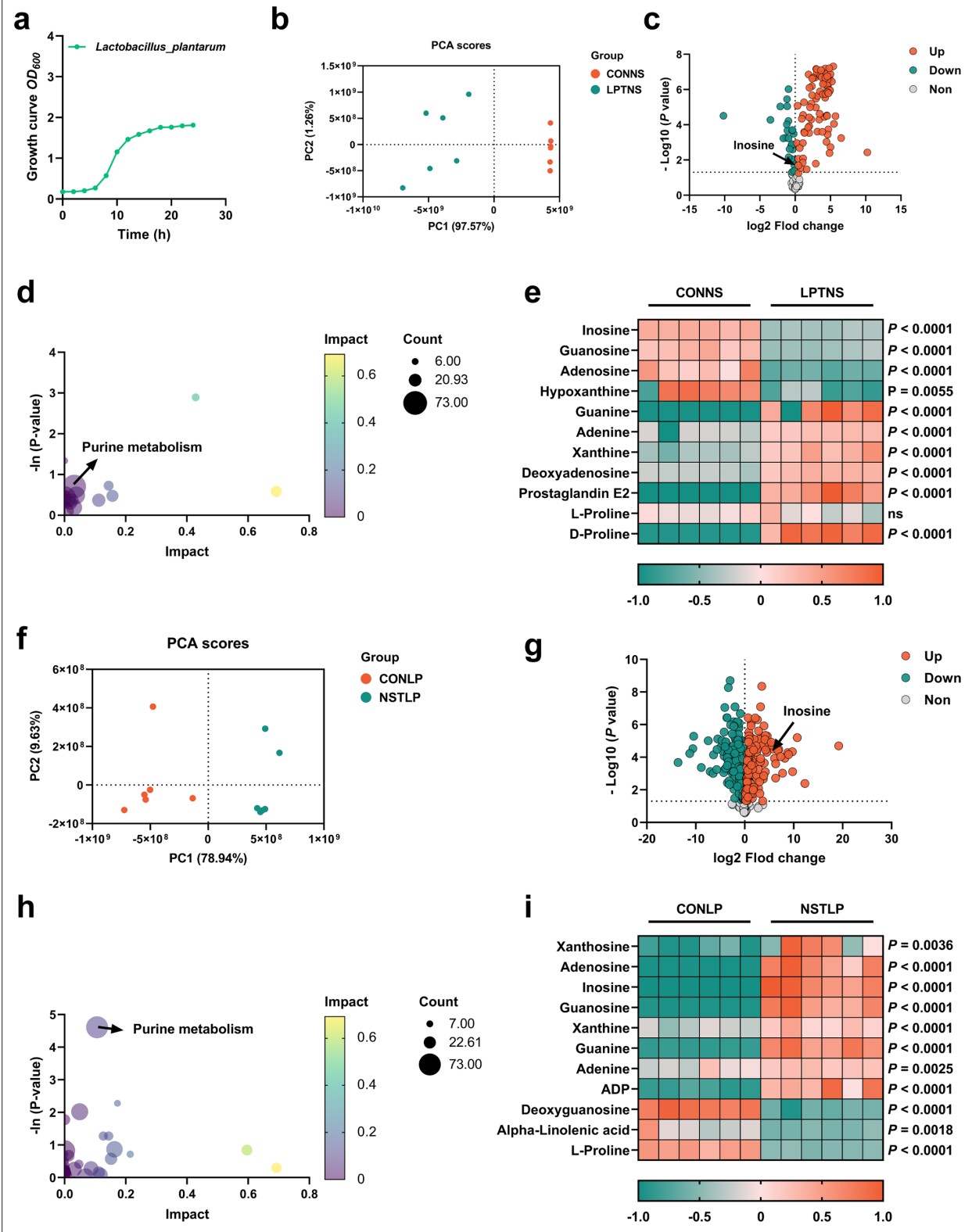

**Figure 4.** Purine metabolism of *L. plantarum* SQ001 was activated by nucleosides. (**a**) Growth curve of *L. plantarum* SQ001. (**b**) Principal component analysis (PCA) score plot of different metabolites collected from nucleoside solution samples with the treatment of LP or CON. All collected data are within the 95% confidence interval. Red represents the CONNS group, green represents the LPTNS group. (**c**) Volcano plots of changed metabolites for the comparison with LPTNS versus CONNS. (**d**) Kyoto encyclopedia of genes and genomes (KEGG)-based pathway over-representation analysis of changed metabolites between CONNS and LPTNS groups. The bubble size represents the percentage of significant metabolites contained in each

*Figure 4 continued on next page*

*Figure 4 continued*

pathway. Bubbles are colored according to the impact. (**e**) Heatmap of extracellular liquid chromatography–mass spectrometry (LC–MS) data showing marker metabolite changes between CONNS and LPTNS groups (*n* = 6). Increases in metabolite levels are shown in red, whereas green indicates decreased metabolite. (**f**) PCA score plot of different metabolites collected from LP samples with the treatment of nucleoside or CON. All collected data are within the 95% confidence interval. Red represents the CONLP group, green represents the NSTLP group. (**g**) Volcano plots of changed metabolites for the comparison with NSTLP versus CONLP. (**h**) KEGG-based pathway over-representation analysis of changed metabolites between NSTLP and CONLP groups. The bubble size represents the percentage of significant metabolites contained in each pathway. Bubbles are colored according to the impact. (**i**) Heatmap of intracellular LC–MS data showing marker metabolite changes between NSTLP and CONLP groups (*n* = 6). Increases in metabolite levels are shown in red, whereas green indicates decreased metabolite. Statistical significance in (**e, i**) was determined by unpaired two-tailed Student's *t*-test. Data with error bars represent mean ± s.e.m. CONNS: control nucleoside solution; LPTNS: *L. plantarum* SQ001-treated nucleoside solution; CONLP: control *L. plantarum* SQ001; NSTLP: nucleoside solution-treated *L. plantarum* SQ001.

## Purine metabolism of *L. plantarum* SQ001 was activated by nucleosides

Although it is clear that *L. plantarum* SQ001 is involved in nucleoside degradation, the specific metabolic impact on *L. plantarum* SQ001 is unclear. Further, after 6 hr of co-culture with nucleoside solutions (inosine, adenosine, and guanosine), principal component analysis (PCA) and volcano plots revealed that *L. plantarum* SQ001 has modified the composition of nucleoside solutions and reduced the amount of inosine present (*Figure 4a–c*). KEGG analysis revealed the enrichment of metabolites in the purine metabolism pathway (*Figure 4d*). Heatmaps of differential metabolites revealed that *L. plantarum* SQ001 significantly reduced levels of inosine (p < 0.0001), guanosine (p < 0.0001), and adenosine (p < 0.0001), while elevating levels of guanine (p < 0.0001), xanthine (p < 0.0001), and proline (p < 0.0001; *Figure 4e*).

We further examined the intracellular metabolic changes in *L. plantarum* SQ001, both with and without incubation with nucleosides (inosine, adenosine, and guanosine). PCA and volcano plots showed that the nucleoside solution altered the intracellular metabolite composition of *L. plantarum* SQ001, with an increase in the content of inosine (*Figure 4f, g*). KEGG analysis indicated a notable enrichment of metabolites in the purine metabolism pathway (*Figure 4h*). Heatmaps illustrating the differences in metabolite levels demonstrated a significant increase in inosine (p < 0.0001), guanosine (p < 0.0001), adenosine (p < 0.0001), guanine (p < 0.0001), xanthine (p < 0.0001), and adenine (p = 0.0025), while alpha-linolenic acid (p = 0.0018) and proline (p < 0.0001) levels showed a significant decrease (*Figure 4i*).

Overall, we have confirmed the absorption and hydrolysis of nucleosides by *L. plantarum* SQ001. Moreover, we have identified several potentially advantageous metabolites, such as linolenic acid and proline that are synthesized. However, the exact mechanism by which *L. plantarum* SQ001 absorbs and hydrolyze nucleosides is unclear.

## *L. plantarum* SQ001 hydrolyze nucleosides through nucleoside hydrolase, which has the potential to alleviate HUA

The genome consisted of 3,549,454 bp and was predicted to contain 3361 coding sequences (CDSs) and 85 RNA genes (*Figure 5a*). A total of 2,739 protein-coding genes in the genome of *L. plantarum* SQ001 were annotated using clusters of orthologous groups (COG). Among these, genes with unknown functions (S) comprised the largest proportion (19.51%), followed by genes involved in (G) carbohydrate transport and metabolism, (E) amino acid transport and metabolism, and (F) nucleotide transport and metabolism (*Figure 5b*).

In the nucleotide transport and metabolism category, the genome of *L. plantarum* SQ001 contained 89 protein-coding genes. Two genes were identified as nucleoside hydrolases (*iunH*), two as ribonucleoside hydrolases (*rihA*, *rihC*), and one as a nucleoside permease (*yxjA*) (*Figure 5c*). These genes may be associated with the alleviation of HUA by *L. plantarum* SQ001. However, the role of *L. plantarum* SQ001 in HUA should be further clarified.

Here, we focused on the role of nucleoside hydrolase (*iunH*) in the degradation of nucleosides by *L. plantarum* SQ001. We first determined the role of *iunH* by heterologous expression, and *E. coli* completely degraded inosine and guanosine after 3 hr following heterologous expression of iunH (*Figure 5d, e* and *Figure 5—figure supplement 1A, B*). We further confirmed through knockout experiments that *L. plantarum* SQ001 exhibited a 50% reduction in nucleoside degradation upon *iunH* gene knockout (*Figure 5f, g* and *Figure 5—figure supplement 1C–F*).

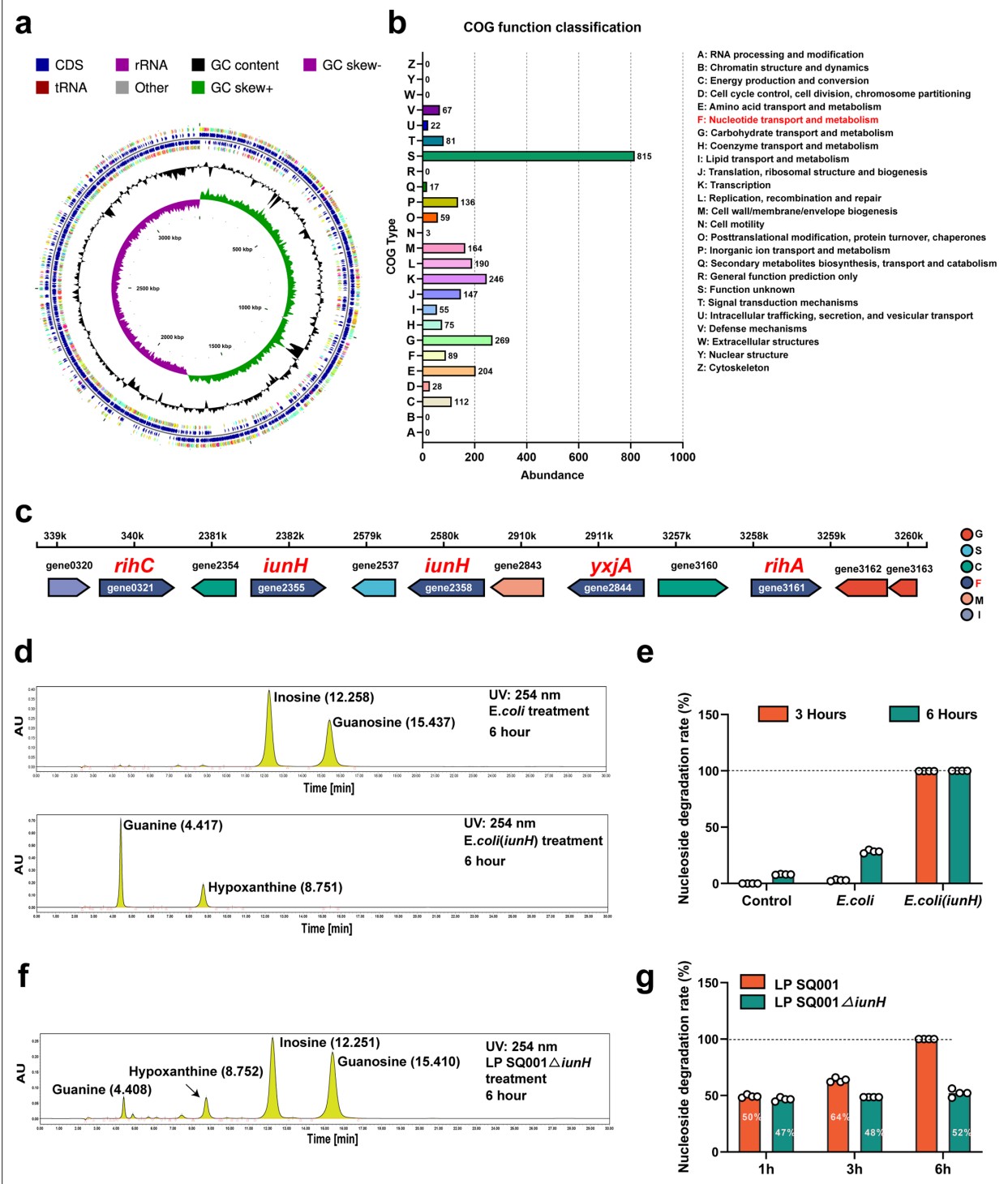

**Figure 5.** *L. plantarum* SQ001 degrades nucleosides through nucleoside hydrolase, which has the potential to alleviate hyperuricemia (HUA). (**a**) Genome map of *L. plantarum* SQ001. (**b**) The clusters of orthologous groups (COG) repertoires of *L. plantarum* SQ001. (**c**) Genes involved in nucleotide transporter metabolism are arranged on each scaffold as a linear plot, with each gene labeled for coding direction and color-coded by COG functional classification. (**d**) Degradation of inosine and guanosine by *E. coli* and *E. coli* (iunH) from high-performance liquid chromatography (HPLC) analysis. (**e**) Nucleoside degradation rate of *L. plantarum* SQ001 gene *iunH* after heterologous expression in *E. coli*. (**f**) Degradation of inosine and guanosine by *L. plantarum* SQ001ΔiunH from HPLC analysis. (**g**) Nucleoside degradation rate of *L. plantarum* SQ001 after knockout of gene *iunH*. yxjA: nucleoside permease; iunH: nucleoside hydrolase; rihC: ribonucleoside hydrolase; rihA: pyrimidine-specific ribonucleoside hydrolase.

The online version of this article includes the following source data and figure supplement(s) for figure 5:

*Figure 5 continued on next page*

*Figure 5 continued*

**Figure supplement 1.** Heterologous expression and knockout validation of the gene iunH.

**Figure supplement 1—source data 1.** Raw unedited gels for *Figure 5—figure supplement 1*.

**Figure supplement 1—source data 2.** Uncropped and labeled gels for *Figure 5—figure supplement 1*.

Overall, we demonstrated that *L. plantarum* SQ001 hydrolyze nucleosides through nucleoside hydrolase *iunH*. However, the mitigating effect of *L. plantarum* SQ001 on hyperuricemic geese remains to be verified.

### *L. plantarum* SQ001 and *L. plantarum* SQ001 with metabolites alleviate HCP diet-induced HUA

To further confirm the functional role of *L. plantarum* SQ001 in HUA, we administered *L. plantarum* SQ001 and *L. plantarum* SQ001 with metabolites to HUA geese via oral gavage for 14 days, respectively (*Figure 6a*). The results showed a markedly improved growth performance, as shown by improved final average weight (p < 0.0001; *Figure 6b*), and decreased mortality (p = 0.0003; *Figure 6c*), relative weight of liver (p = 0.0800) and kidney (p = 0.0200; *Figure 6d*). Additionally, both *L. plantarum* SQ001 and *L. plantarum* SQ001 with metabolites significantly reduced serum UA (p < 0.0001), XOD (p < 0.0001), BUN (p < 0.0001), and creatinine contents (p < 0.0001), and decreased serum Interleukin-1β (IL-1β, p < 0.0001), Tumor necrosis factor-α (TNF-α, p < 0.0001), and Interferon-γ (IFN-γ, p < 0.0001) levels (*Figure 6e–g*). Geese oral with the *L. plantarum* SQ001 and *L. plantarum* SQ001 with metabolites showed attenuated jejunum phenotypes damage including villus length (p = 0.0051), and crypt depth (p = 0.0122; *Figure 7a, b*). In addition, *L. plantarum* SQ001 relieved pathological injuries such as inflammatory cell infiltration in the liver and kidney as reflected by the pathological sections (p < 0.0001; *Figure 7c, d*). These findings demonstrate the alleviation of HUA in geese by *L. plantarum* SQ001, but the role played by gut microbes is unclear.

Sequencing analysis of cecal chyme revealed that both *L. plantarum* SQ001 and *L. plantarum* SQ001 with metabolites significantly increased microbiota richness (Chao index, p < 0.0001) and uniformity (Shannon index, p < 0.0001) compared to the HUA group (*Figure 8a* and *Figure 8—figure supplement 1A*). PCoA analysis showed the separation of gut microbiota in *L. plantarum* SQ001, *L. plantarum* SQ001 with metabolites, and HUA groups (*Figure 8b*). Sequencing results showed that *L. plantarum* SQ001 and *L. plantarum* SQ001 with metabolites significantly enhanced the relative abundance of phylum *Firmicutes* (*Figure 8—figure supplement 1B*), family *Lachnospiraceae* (p = 0.0218), family *Ruminococcaceae* (p = 0.0049; *Figure 8c*), family *Lactobacillaceae* (p = 0.0005; *Figure 8d*), and family *Butyricicoccaceae* (p = 0.0278; *Figure 8e* and *Figure 8—figure supplement 1C*). Meanwhile, the relative abundance of genus *Lactiplantibacillus* (p = 0.0012), *Butyricicoccus* (p = 0.0350), and species *L. plantarum* (p = 0.0003), *B. pullicaecorum* (p = 0.0059) were also markedly boosted (*Figure 8d, e* and *Figure 8—figure supplement 1D*). Overall, *L. plantarum* SQ001 ameliorated HCP diet-induced gut microbiological disturbances, restored and increased the abundance of *Lactobacillus* spp. and *Butyricicoccus* spp. and ultimately alleviated HUA in geese.

### *L. plantarum* SQ001 alleviates HUA in a mice model

To further explore the role of *L. plantarum* SQ001 in alleviating HUA, Kunming mice were utilized as an additional HUA model, and oral gavage of *L. plantarum* SQ001 was administered (*Figure 9a*). *L. plantarum* SQ001 significantly reduced knee joint thickness (p = 0.0011; *Figure 9b*), relative kidney weight (p = 0.0038; *Figure 9c*), serum UA levels (p = 0.0022), and serum XOD activity (p = 0.0348) in the HUA model mice (*Figure 9d*). Further analysis of key hepatic enzymes involved in UA formation revealed that *L. plantarum* SQ001 significantly lowered the expression of phosphoribosyl pyrophosphate synthetase (PRPS, p = 0.0044) and xanthine oxidase (XO, p < 0.0001; *Figure 9e*) in the liver. Additionally, *L. plantarum* SQ001 reduced the protein expression associated with UA reabsorption in the kidneys, including glucose transporter 9 (GLUT9, p = 0.0085), and increased the expression of the urate excretion protein ATP-binding cassette transporter G2 (ABCG2, p = 0.1028; *Figure 9f*). Collectively, these findings suggest that *L. plantarum* SQ001 can alleviate HUA, primarily by decreasing UA production in the liver and enhancing UA excretion in the kidneys.

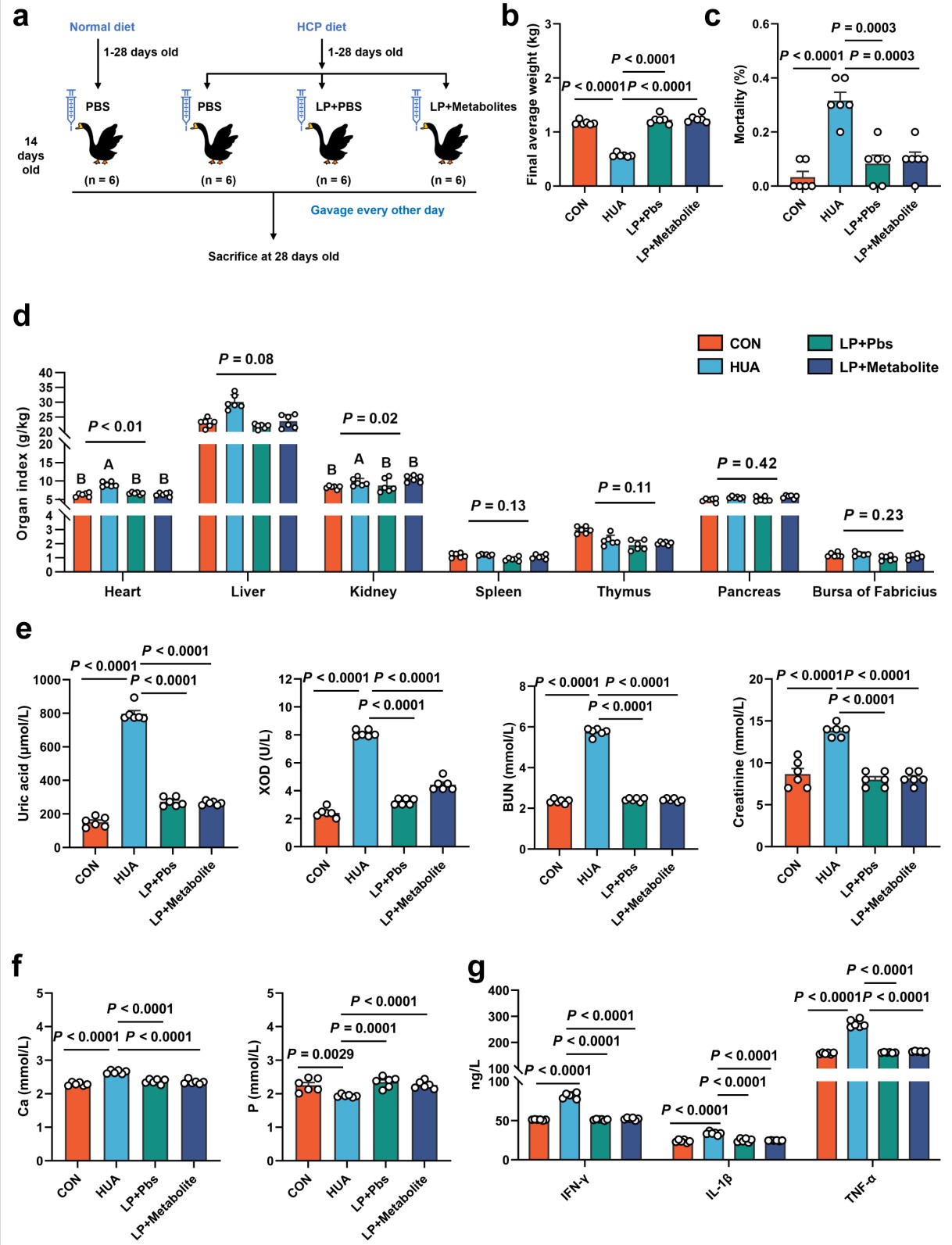

**Figure 6.** *L. plantarum* SQ001 and *L. plantarum* SQ001 with metabolites alleviate HCP diet-induced HUA. (**a**) Experimental design. (**b**) Effect of *L. plantarum* SQ001 and *L. plantarum* SQ001 with metabolites on the final average weight in HCP diet-treated geese (*n* = 6). (**c**) Effect of *L. plantarum* SQ001 and *L. plantarum* SQ001 with metabolites on the mortality in HCP diet-treated geese (*n* = 6). (**d**) Effect of *L. plantarum* SQ001 and *L. plantarum* SQ001 with metabolites on the organ index in HCP diet-treated geese (*n* = 6). (**e**) Effect of *L. plantarum* SQ001 and *L. plantarum* SQ001 with metabolites

*Figure 6 continued on next page*

*Figure 6 continued*

on the serum uric acid, xanthine oxidase (XOD), blood urea nitrogen (BUN), and creatinine in HCP diet-treated geese ($n = 6$). (**f**) Effect of *L. plantarum* SQ001 and *L. plantarum* SQ001 with metabolites on the serum Ca and P levels in HCP diet-treated geese ($n = 6$). (**g**) Effect of *L. plantarum* SQ001 and *L. plantarum* SQ001 with metabolites on the serum IFN-γ, IL-1β, and TNF-α levels in HCP diet-treated geese ($n = 6$). Statistical significance in (**b–f**) was determined by unpaired two-tailed Student's *t*-test. Significant differences in (**d**) were evaluated by one-way ANOVA with Bonferroni's multiple comparisons test. Data with error bars represent mean ± s.e.m.

To further confirm the HUA-alleviating effect of *L. plantarum* SQ001, we performed an untargeted metabolomic analysis of serum metabolites. PCA demonstrated that the two groups exhibited distinct metabolite profiles (*Figure 10a, b*). Pathway enrichment analysis revealed that the identified metabolites were predominantly linked to purine metabolism (*Figure 10c, d*). *L. plantarum* SQ001 decreased the levels of inosine and xanthosine, and significantly increased the L-proline, L-valine, L-arginine, L-targinine, and L-methionine levels (*Figure 10e*). Further investigation of the gut microbiota showed that *L. plantarum* SQ001 significantly increased the Chao ($p < 0.0001$) and Shannon indices ($p = 0.0002$; *Figure 10f*). PCA revealed distinct differences in the gut microbiota among the four groups (*Figure 10g*). Operational taxonomic unit species composition analysis revealed that *L. plantarum* SQ001 restored the ratio of *Fimicutes* to *Bacteroidota*, increased the abundance of *Lactobacillaceae*, *Lactobacillus*, *Lactobacillus murinus*, and *Lactobacillus reuteri*, and decreased the abundance of *Staphylococcus* and *Staphylococcus xylosus* (*Figure 10h–k*). Moreover, *L. plantarum* SQ001 increased the abundance of *L. plantarum* (*Figure 10l*). Additionally, Spearman correlation analysis showed that *Lactobacillus* ($p = 0.0360$) and *L. plantarum* ($p = 0.0120$) had a significantly negative association with serum inosine level, while *Bacteroides* showed a substantially positive correlation with inosine ($p = 0.0070$; *Figure 10m*). Furthermore, the analysis indicated that *Lactobacillus* and *L. plantarum* had a substantially positive correlation with serum indole-3-carboxylic acid (ICA) level, while *Bacteroides* exhibited a significantly negative correlation with ICA.

Overall, these results demonstrated that *L. plantarum* SQ001 regulates hepatic and renal UA metabolism by increasing the abundance of beneficial gut bacteria, such as *Lactobacillus*, and reducing the levels of UA precursors.

## Discussion

Gut microbial disorders may be strongly associated with the formation of HUA induced by high purine diets. In our study, we observed a significant decrease in the abundance of *Lactobacillus*, *L. plantarum*, *Butyricicoccus*, and *B. pullicaecorum* in the cecum of HUA geese, and a significant increase in the abundance of *R. torques* and *R. gauvreauil* groups. *L. plantarum* SQ001, derived from HUA model geese, showed nucleoside degradation function through an in vitro assay. Following oral gavage in HUA geese significantly reduced serum UA levels and alleviated hepatic and renal inflammation, simultaneously restoring intestinal microbial disorders. In addition, oral gavage of *L. plantarum* SQ001 to an HUA mouse model similarly increased *Lactobacillus* and *L. plantarum* abundance, and reduced the serum UA level (*Figure 11*). These results suggested that host-derived beneficial bacteria, *L. plantarum*, improved the intestinal flora and alleviated the HUA in host.

The low abundance of *Lactobacillus* spp. and *Butyricicoccus* spp. suggests a possible antagonistic role in the development of HUA. Studies have shown that oral administration of *Lactobacillus* spp. such as *L. plantarum*, *Lactobacillus rhamnosus*, and *Limosilactobacillus* could maintain gut microbiota balance, increase the abundance of *Lactobacillus*, and reduce serum UA levels (*Wang et al., 2022*; *Cao et al., 2022a*; *Zhao et al., 2022a*). Our findings also showed that *L. plantarum* SQ001 ameliorated the intestinal microbiota disturbance and serum UA elevation, and restored the *Firmicutes* to *Bacteroidetes* ratio, which are associated with metabolic diseases. In HUA mouse model, *L. plantarum* SQ001 reduced the expression of hepatic UA synthesis proteins PRPS, XO, and renal UA reabsorption protein GLUT9, while increasing the expression of renal UA excretory protein ABCG2. In a healthy body, UA levels are regulated by a delicate balance between its production and excretion. Excessive production or insufficient excretion of UA can lead to the development of HUA (*Li et al., 2023b*). The enzymes PRPS and XO are crucial for UA synthesis, catalyzing its production. GLUT9 is located on the apical brush-border membrane (urine side) of renal proximal tubules and functions as a UA reabsorption transporter. Conversely, ABCG2 is primarily found on the basolateral (blood-side) membrane of

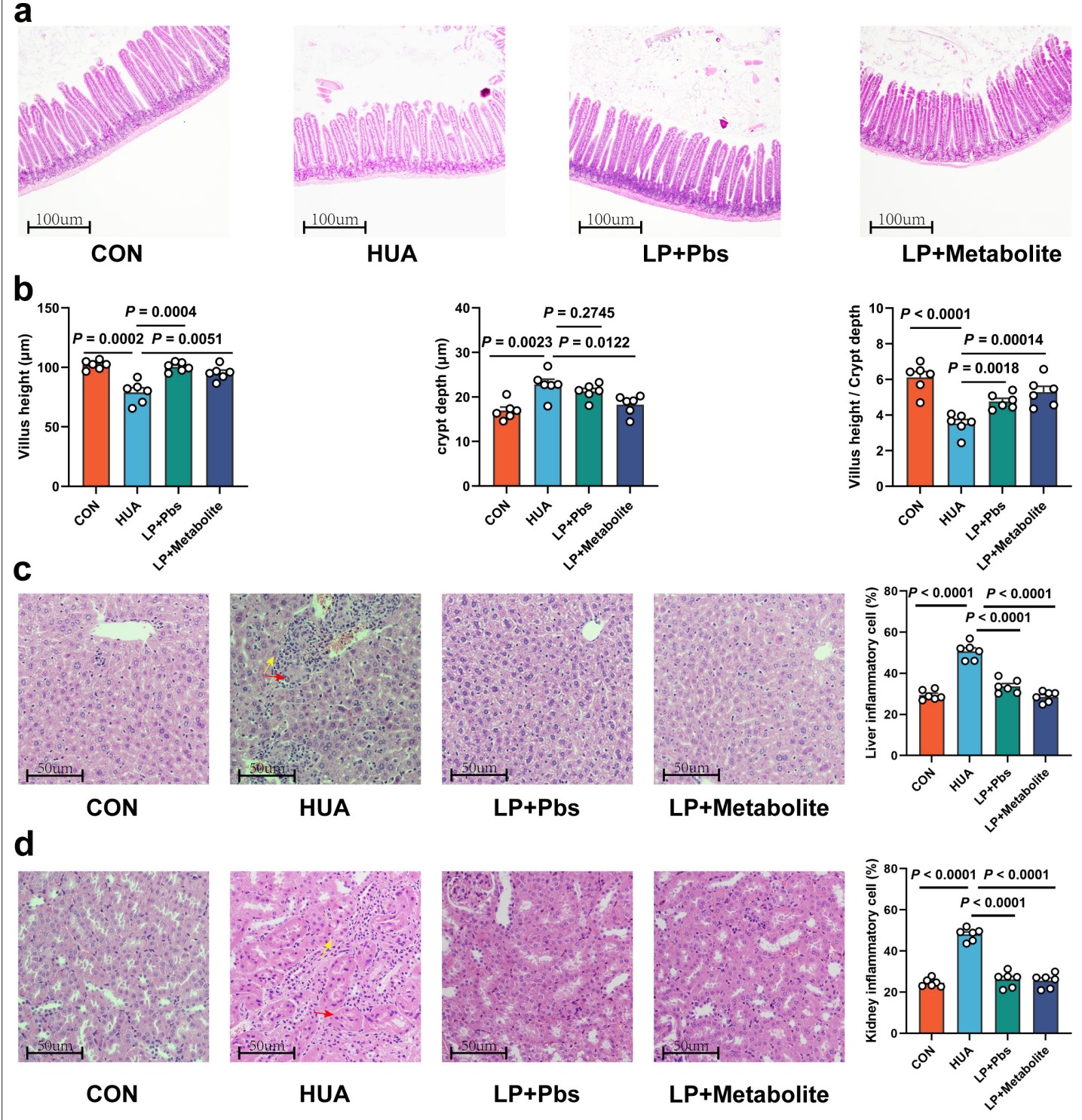

**Figure 7.** *L. plantarum* SQ001 and *L. plantarum* SQ001 with metabolites could alleviate HCP diet-induced damage in the jejunum, liver, and kidneys. (**a**) Representative H&E staining images and a histologic score of the jejunum section of geese (*n* = 6). Scale bar: 100 μm. (**b**) Villi height, crypt depth, and the value of villi height/crypt depth (*n* = 6). Six crypts were counted for each section. (**c**) Representative H&E staining images and inflammatory cell rate of the liver section of geese (*n* = 6). Scale bar: 50 μm. (**d**) Representative H&E staining images and inflammatory cell rate of kidney section of geese (*n* = 6). Scale bar: 50 μm. Statistical significance in b was determined by unpaired two-tailed Student's *t*-test. Data with error bars represent mean ± s.e.m.

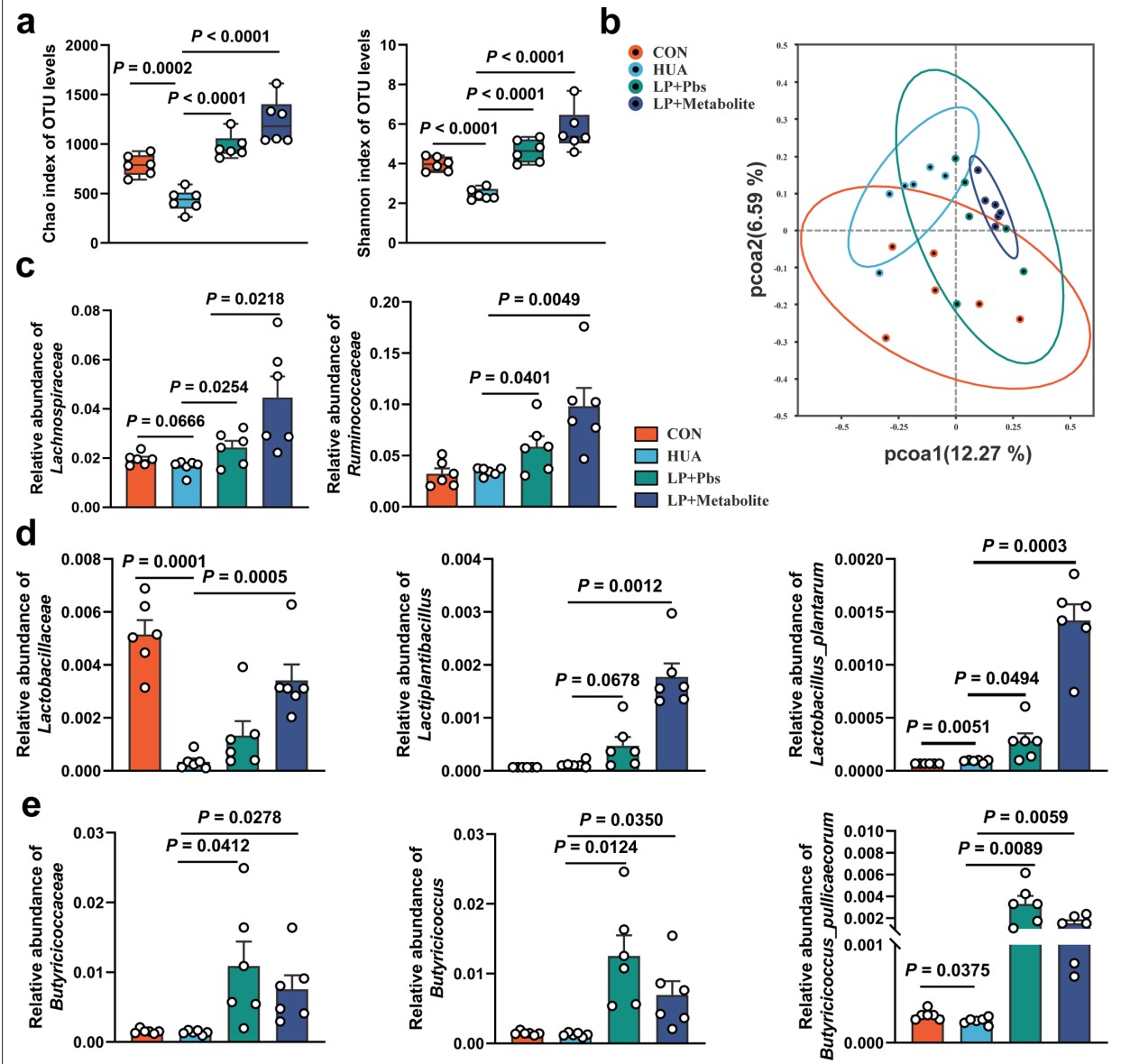

**Figure 8.** *L. plantarum* SQ001 and *L. plantarum* SQ001 with metabolites alleviate HCP diet-induced gut microbial disorder. (**a**) Chao and Shannon indices of indicated groups based on alpha diversity analysis (*n* = 6). (**b**) Principal component analysis of bacteria with 95% confidence regions of indicated groups (*n* = 6). (**c**) Effect of *L. plantarum* SQ001 and *L. plantarum* SQ001 with metabolites on the relative abundance of family *Lachnospiraceae* and *Ruminococcaceae* in HCP diet-treated geese (*n* = 6). (**d**) Effect of *L. plantarum* SQ001 and *L. plantarum* SQ001 with metabolites on the relative abundance of family *Lactobacillaceae*, genus *Lactiplantibacillus*, and species *Lactobacillus plantarum* in HCP diet-treated geese (*n* = 6). (**e**) Effect of *L. plantarum* SQ001 and *L. plantarum* SQ001 with metabolites on the relative abundance of family *Butyricicoccaceae*, genus *Butyricicoccus*, and species *Butyricicoccus pullicaecorum* in HCP diet-treated geese (*n* = 6). Statistical significance in (**a, c–e**) was determined by unpaired two-tailed Student's *t*-test. Data with error bars represent mean ± s.e.m.

The online version of this article includes the following figure supplement(s) for figure 8:

**Figure supplement 1.** Effect of *L. plantarum* SQ001 and *L. plantarum* SQ001 with metabolites on gut microbiota in geese.

proximal tubule cells in the kidney, facilitating the outward excretion of UA from the renal tubules. The downregulation of GLUT9 expression and the upregulation of ABCG2 expression facilitate increased UA excretion, thereby reducing serum UA accumulation (*Dalbeth et al., 2021*).

In this study, *L. plantarum* SQ001 further reduced the expression of the pro-inflammatory cytokine IL-1β in serum. Meanwhile, liver and kidney inflammation in HUA geese were alleviated. However, the mechanism by which *L. plantarum* SQ001 alleviated inflammation has not been elucidated.

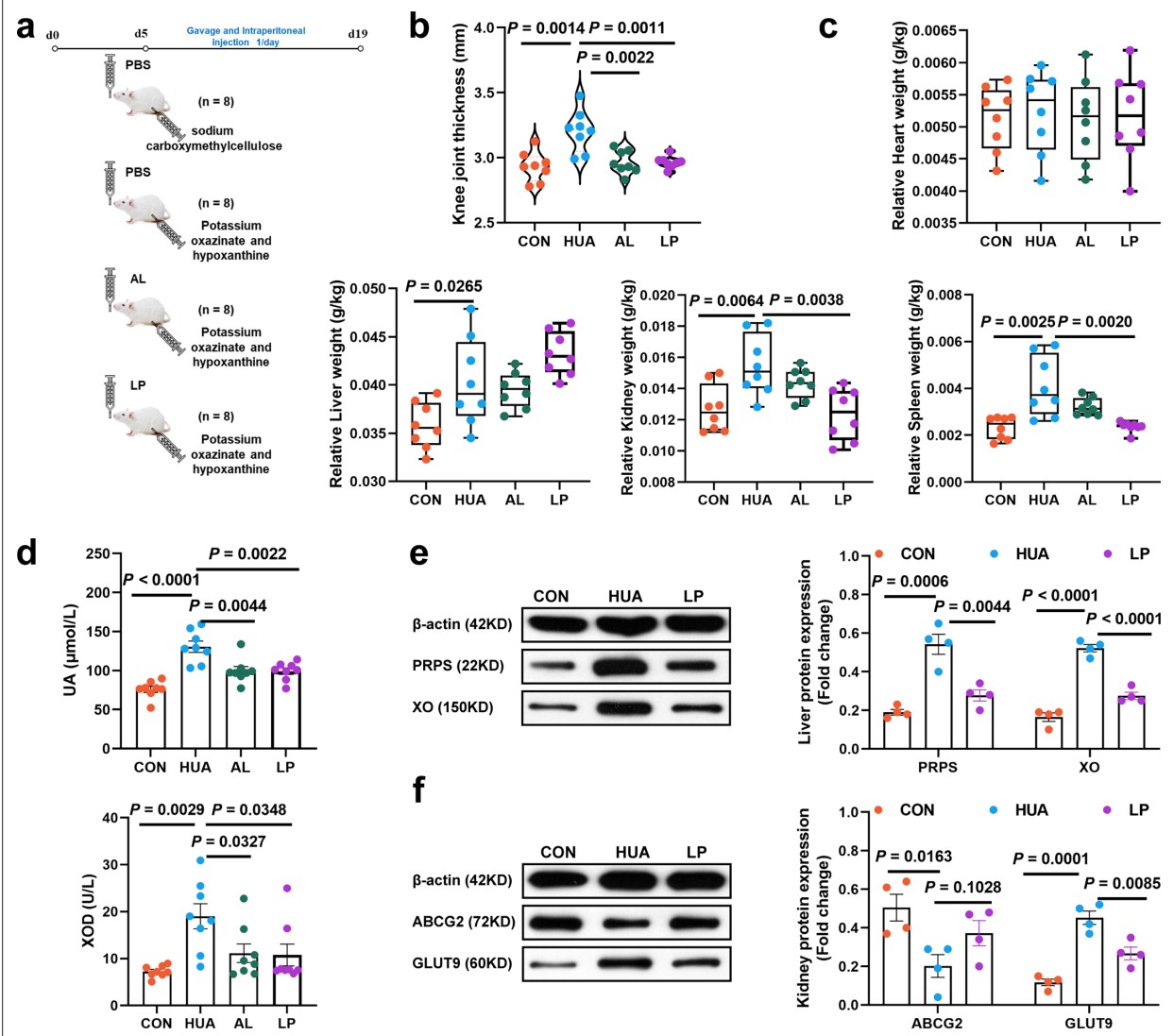

**Figure 9.** Effect of *L. plantarum* SQ001 on alleviating HUA and protein expression levels of uric acid (UA) metabolism in mice. (**a**) Experimental design. (**b**) Effect of *L. plantarum* SQ001 on the knee joint thickness in mice (*n* = 8). (**c**) Effect of *L. plantarum* SQ001 on the organ index (heart, liver, kidney, and spleen) in mice (*n* = 8). (**d**) Effect of *L. plantarum* SQ001 on the serum UA, xanthine oxidase (XOD) in HCP diet-treated geese (*n* = 8). (**e**) Representative western blotting images and quantification of proteins (phosphoribosyl pyrophosphate synthetase [PRPS], XO) in the liver tissue between the CON, HUA, and LP groups (*n* = 4). (**f**) Representative western blotting images and quantification of proteins (ABCG2, GLUT9) in the kidney tissue between the CON, HUA, and LP groups (*n* = 4). Statistical significance in (**b–f**) was determined by unpaired two-tailed Student's *t*-test. Data with error bars represent mean ± s.e.m. CON, control group; HUA, hyperuricemia model group; AL, allopurinol group; LP, *L. plantarum* SQ001 group.

The online version of this article includes the following source data for figure 9:

**Source data 1.** Raw unedited gels for *Figure 9*.

**Source data 2.** Uncropped and labeled gels for *Figure 9*.

Studies have shown that *L. plantarum* could alleviate inflammation by promoting short-chain fatty acids (SCFAs) production and the abundance of SCFA-producing beneficial bacteria (*Chien et al., 2022*; *Wei et al., 2021*). *Butyricicoccus* and *B. pullicaecorum* have been reported to enhance intestinal barrier function and inhibit inflammation by improving butyrate production, which indicated the probable underlying mechanism (*Marteau, 2013*; *Devriese et al., 2017*). We observed that *L. plantarum* SQ001 enhanced the abundance of *Lactobacillus*, *L. plantarum*, *Butyricicoccus*, and *B. pullicaecorum* in the gut, improved intestinal villus morphology and restored the intestinal barrier. *R. torques group* has been associated with the development of gastrointestinal disorders, and its abundance was significantly increased in patients with Crohn's disease and inflammatory bowel disease

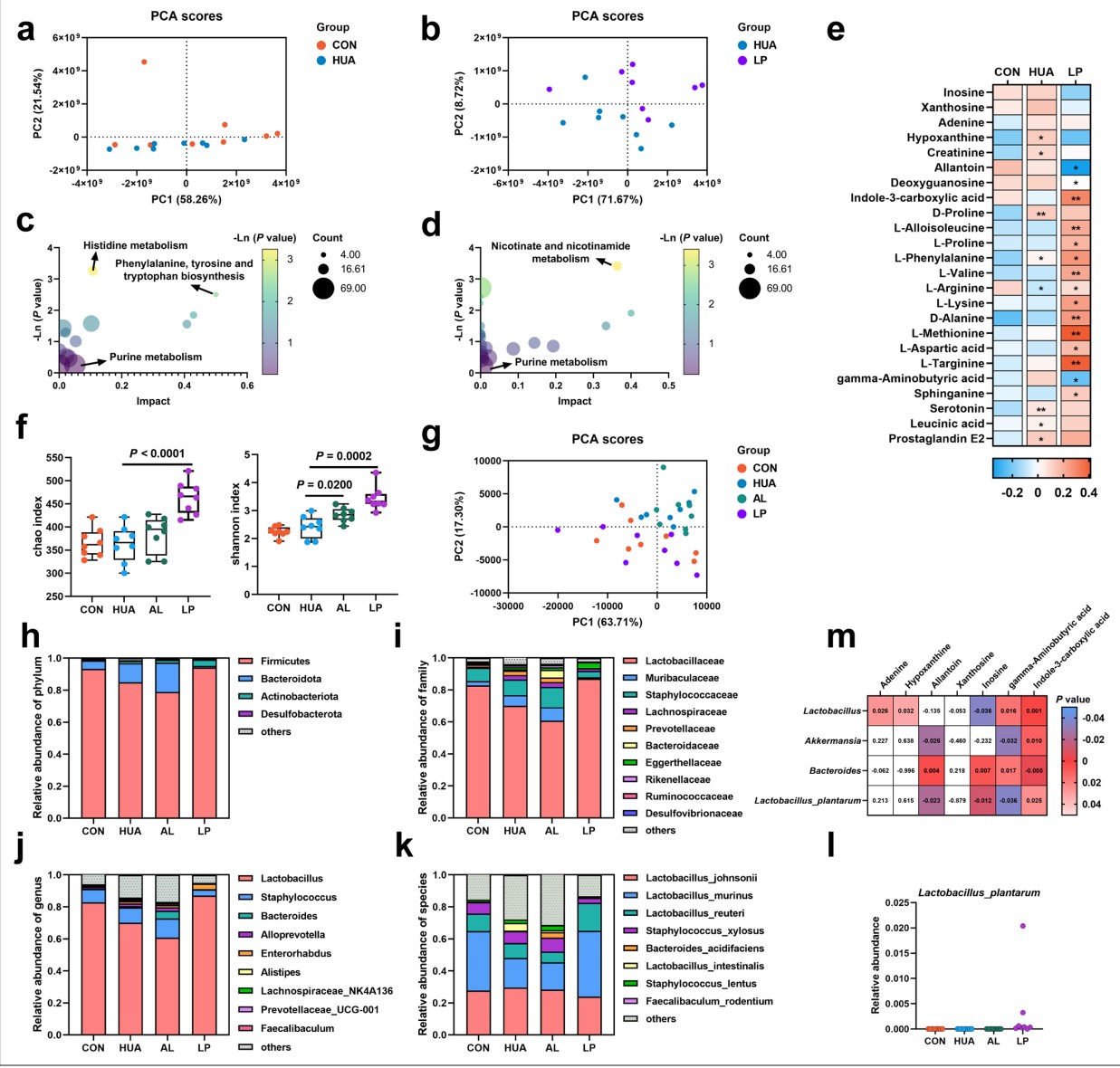

**Figure 10.** Effect of *L. plantarum* SQ001 on serum metabolites and gut microbiota in mice. (**a**) Principal component analysis of the serum samples. The yellow color represents the CON group (*n* = 8), while the green color represents HUA group (*n* = 8). (**b**) Principal component analysis of the serum samples. The green represents the HUA group (*n* = 8), while the purple represents LP group (*n* = 8). (**c**) Kyoto encyclopedia of genes and genomes (KEGG)-based pathway over-representation analysis of changed metabolites between CON and HUA groups. The bubble size represents the percentage of significant metabolites contained in each pathway. Bubbles are colored according to the impact. (**d**) KEGG-based pathway over-representation analysis of changed metabolites between HUA and LP groups. The bubble size represents the percentage of significant metabolites contained in each pathway. Bubbles are colored according to the impact. (**e**) Heatmap of serum metabolites liquid chromatography–mass spectrometry (LC–MS) data showing marker metabolite changes between CON, HUA, and LP groups (*n* = 8). Increases in metabolite levels are shown in red, whereas blue indicates decreased metabolite. (**f**) Chao and Shannon indices of indicated groups based on alpha diversity analysis (*n* = 8). (**g**) Principal component analysis of bacteria with 95% confidence regions of indicated groups (*n* = 8). Relative abundance of gut microbes at the phylum (**h**), family (**i**), genus (**j**), and species (**k**) levels of mice (*n* = 8). (**l**) Effect of *L. plantarum* SQ001 on the relative abundance of species *Lactobacillus plantarum* in mice (*n* = 7). (**m**) Spearman correlation analysis between serum metabolites and gut microbes. Statistical significance in (**e, f**) was determined by unpaired two-tailed Student's *t*-test. ns, no significance. *p < 0.05, **p < 0.01, ***p < 0.001. Data with error bars represent mean ± s.e.m. CON, control group; HUA, hyperuricemia model group; AL, allopurinol group; LP, *L. plantarum* SQ001 group.

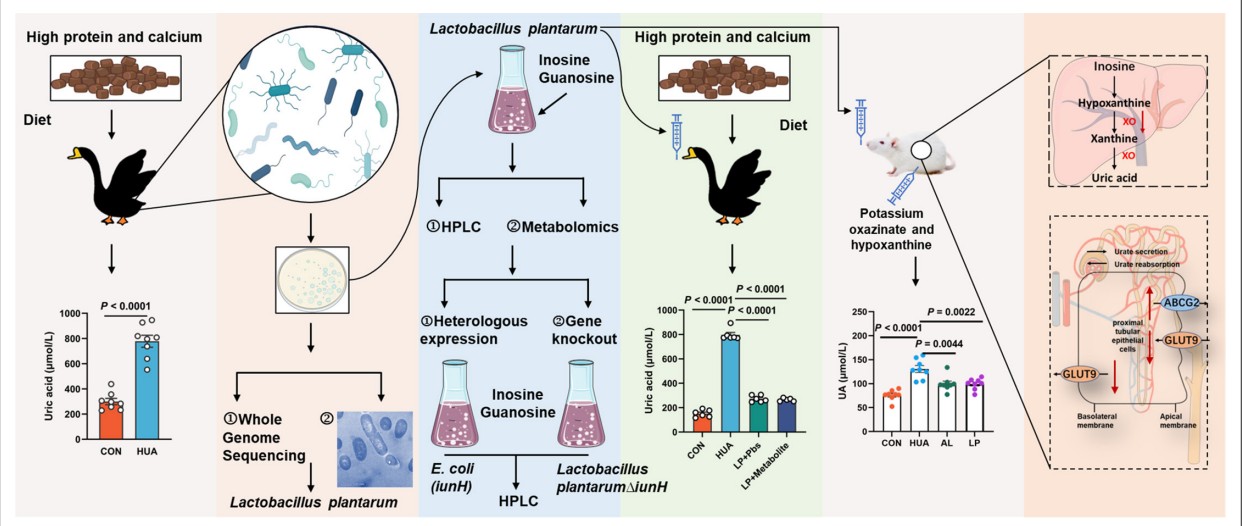

**Figure 11.** Working hypothesis. The results suggest that host-derived *Lactobacillus plantarum* SQ001 hydrolyzes nucleosides through the nucleoside hydrolase iunH, reducing intestinal nucleoside transport, lowering hepatic uric acid production, restoring renal excretion, and ultimately alleviating hyperuricemia (HUA). *iunH*: Inosine-uridine nucleoside N-ribohydrolase; GLUT9: glucose transporter 9, UA reabsorption transporter; ABCG2: ATP-binding cassette transporter G2, UA excretion transporter.

(*Joossens et al., 2010*; *Henke et al., 2019*). *R. gauvreauil* group is associated with low gut microbial abundance and its abundance is higher in patients with atherosclerotic cardiovascular disease (*Henke et al., 2019*). However, there is insufficient evidence for a causal relationship between *R. torques* group, *R. gauvreauil* group, and the development of HUA. Our results suggested that *L. plantarum* SQ001 reversed the HCP diet-induced increase in abundance of *R. torques* and *R. gauvreauil* groups. In addition, in the HUA mouse model, *L. plantarum* SQ001 increased the abundance of *Lactobacillaceae*, *Lactobacillus*, *L. murinus*, and *L. reuteri*, while decreasing the abundance of *Staphylococcus* and *S. xylosus*. *S. xylosus* is a commensal pathogen on the skin and is one of the causative agents of skin inflammation, but the role of *S. xylosus* in the gut is unclear (*Kim et al., 2017*). The findings also indicated a robust positive correlation between *Lactobacillus*, especially *L. plantarum*, and the serum ICA levels. Research has demonstrated that toxins are metabolized by enzymes like Cytochrome P450 2E1 (CYP2E1) and converted into reactive species, causing oxidative damage and liver failure. ICA binds to and deactivates CYP2E1, thereby inhibiting the formation of harmful reactive intermediates and reducing oxidative damage to hepatocytes (*Li et al., 2024*). Moreover, *L. gallinarum*-derived ICA suppressed colorectal cancer by inhibiting CD4[+] Treg differentiation and enhancing CD8[+] T cell function (*Fong et al., 2023*). However, the mitigating effects and mechanisms of ICA in HUA need further validation. In conclusion, *L. plantarum* SQ001 improved the intestinal microbiota disorders, increased the abundance of beneficial bacteria, and alleviated HUA. Impressively, Spearman correlation analysis revealed a significant negative association between inosine and both *Lactobacillus* and *L. plantarum* in terms of serum metabolites and gut microbes. *Lactobacillus* and *L. plantarum* may work by hydrolyzing inosine.

Previous studies have shown that some *Lactobacillus* can alleviate HUA. The strain *Limosilactobacillus fermentum* JL-3, isolated from Chinese slurry water, degraded UA and alleviated HUA, but the specific UA-lowering mechanism has not been elucidated (*Cao et al., 2022b*). Apart from hydrolyzing UA, *Lactobacillus* spp. from various sources could lower serum UA levels by hydrolyzing nucleosides. For instance, *Lactobacillus paracasei* X11 and *L. plantarum* DM9218-A, isolated from Chinese sauerkraut, and *L. plantarum* GKM3, isolated from Chinese mustard, have shown this capability (*Li et al., 2014*; *Hsu et al., 2019*; *Li et al., 2023a*; *Li et al., 2023b*). However, the mechanism by which these *Lactobacillus* spp. degrade nucleosides has not yet been fully understood. Recent studies have shown that *L. plantarum* from Chinese sauerkraut and *Lactobacillus aviarius* from chicken cecum contents are capable of degrading nucleosides, and the possible degradation mechanism was elucidated by gene cloning and recombination to obtain ribonucleoside hydrolase RihA-C and nucleoside hydrolase Nhy69 (*Li et al., 2023a*; *Li et al., 2023b*). Even though the validation of degradation function

in bacteria has been investigated in these studies, functional verification on the deficiency of critical genes in strains has not been elucidated. Additionally, the investigation of host-derived and repurposed *Lactobacillus* strains with HUA-relieving properties remains unexplored, while these particular *Lactobacillus* strains often exhibit antagonistic effects within the host suffering from HUA. Our results showed that *L. plantarum* SQ001 contains four nucleoside hydrolysis-related enzymes (*iunH*, *yxjA*, *rihA*, and *rihC*), and suggested that nucleoside hydrolase *iunH* is involved in nucleosides (inosine and guanosine) degradation. Metabolomics showed that nucleosides were efficiently converted to nucleobases in this assay. Although a direct link between nucleobase transport and host urate levels has not yet been established, the absence of nucleobase transport proteins in intestinal epithelial cells may be necessary for reducing their uptake and metabolism of urate production, resulting in a reduction in host serum urate. Moreover, changes in intracellular and extracellular ectonucleoside levels, which may reveal a role for nucleoside transporter-related enzymes, such as *yxjA*, but further validation is needed. Metabolomics indicated an increase in linolenic acid and proline after co-culture with nucleosides. Studies suggest that consuming linolenic acid-rich foods is linked to a reduced risk of gout recurrence, and alpha-linolenic acid inhibits the URAT1 protein, which reabsorbs UA in the kidneys (*Zhang et al., 2019b*; *Saito et al., 2020*). Additionally, microbial-derived proline enhances UA excretion in the intestines and kidneys, helping to alleviate HUA (*Fu et al., 2024*).

In conclusion, our study investigated the changes in gut microbes of HUA geese and mouse, and found the critical genes with nucleoside hydrolysis function in host-derived *L. plantarum* strain, finally demonstrating its purine nucleoside hydrolysis function in vitro and in vivo experiments. This study is expected to contribute insight into HUA metabolism and provide new strategies for HUA or gout therapy via host-derived gut microbe.

# Materials and methods

## Key resources table

| Reagent type (species) or resource | Designation | Source or reference | Identifiers | Additional information |
|---|---|---|---|---|
| Strain, strain background (*Lactobacillus plantarum*) | SQ001 | Lab stock | CGMCC NO 24791 | |
| Strain, strain background (*E. coli*) | GB2005 | Lab stock | | (HS996, ΔrecET, ΔybcC). The endogenous recET locus and the DLP12 prophage ybcC, which encodes a putative exonuclease similar to the Redα, were deleted |
| Strain, strain background (*E. coli*) | GB05-dir | Lab stock | | (GB2005, araC-BAD-ETgA) recE, recT, redB3; and recA under BAD promoter was inserted at the ybcC locus |
| Strain, strain background (*E. coli*) | Nissle1917/p15A-cm-Pgenta | This study | | The Pgenta promoter gene was heterologous expressed in *E. coli* Nissle1917 |
| Strain, strain background (*E. coli*) | Nissle1917/p15A-cm-iunH | This study | | The iunH gene was heterologous expressed in *E. coli* Nissle1917 |
| Antibody | anti-GLUT9 (Rabbit monoclonal) | Proteintech | Cat#26486-1-AP; RRID: AB_2880532 | 1:1000 |
| Antibody | anti-ABCG2 (Rabbit monoclonal) | Abcam | Cat#ab108312; RRID: AB_10861951 | 1:5000 |
| Antibody | anti-PRRS (Mouse monoclonal) | Bioss | Cat#bs-4504R; RRID: AB_11063921 | 1:1000 |
| Antibody | anti-XO (Mouse monoclonal) | Abcam | Cat#ab109235; RRID: AB_10863199 | 1:5000 |

*Continued on next page*

*Continued*

| Reagent type (species) or resource | Designation | Source or reference | Identifiers | Additional information |
|---|---|---|---|---|
| Antibody | Anti-β-actin (Mouse monoclonal) | Proteintech | Cat#60009-1-Ig; RRID: AB_2154438 | 1:5000 |
| Antibody | HRP goat anti-mouse IgG (goat polyclonal) | proteintech | Cat#SA00001-1; RRID: AB_2722565 | 1:5000 |
| Antibody | HRP goat anti-rabbit IgG (goat polyclonal) | proteintech | Cat#SA00001-2; RRID: AB_2722564 | 1:6000 |
| Sequence-based reagent | p15A-5 | This paper | PCR primers | AAACTACCGCATTAAAGCTT |
| Sequence-based reagent | p15A-3 | This paper | PCR primers | CTGAACCGACGACCGGGTCG |
| Sequence-based reagent | p15A-Pgenta-5 | This paper | PCR primers | CAGAAATTCGAAAGCAAATTCGACCCGGTCGTCGGTTC AGGAAGGCACGAACCCAGTTG |
| Sequence-based reagent | genta-60kDa-3 | This paper | PCR primers | CGTTGCTGCTCCATAACAT |
| Sequence-based reagent | Pgenta-iunH-5 | This paper | PCR primers | TACGCCGTGGGTCGATGTTTGATGTTATGGAGCAGCAAC GGAGGATGTTTTACATGGCA |
| Sequence-based reagent | iunH-3 | This paper | PCR primers | TTTGACAGCTTATCATCGATAAGCTTTAATGCGGTAGTT TCTAATGGGCTTTAAATAAT |
| Sequence-based reagent | erm-5 | This paper | PCR primers | GTTCTATGCTTTCTTTTTGTAGCCGGCTAAACGGATA GTCCCCCAAAATCATCTTGCCTTTGATATTGAGGTATCATTT |
| Sequence-based reagent | erm-3 | This paper | PCR primers | CCTTTTTAATCACAATTCAGAAAATATCATAATATC TCATTTCACTAAATAATAGTGAACTTAGCCGTTAAATATTTTA |
| Sequence-based reagent | sacB-5 | This paper | PCR primers | GTTCACTATTATTTAGTGAA |
| Sequence-based reagent | sacB-3 | This paper | PCR primers | AGTGTGACTCTAGTAGAGAGCGTTCACCGACAA ACAACAGTTTGTTAACTGTTAATTGT |
| Sequence-based reagent | Pi-HAR-2 | This paper | PCR primers | GACATTCCCGGCAATCGGAC |
| Sequence-based reagent | Pi-HAR-1 | This paper | PCR primers | AAGGCAAGATGATTTTGGGG |
| Sequence-based reagent | Pi-HAL-2 | This paper | PCR primers | TTTGAACACTCATGTTTAAC |
| Sequence-based reagent | Pi-HAL-1 | This paper | PCR primers | GATGGATGTTCAACTAATTTGTCCGATTGCCG GGAATGTCGCTGATTGATCTGAAAGGA |
| Sequence-based reagent | P15A-PiHAR-2 | This paper | PCR primers | TGTTAGTCATTTTCTTTCTCAACCTCGTCATTGGCACCAAG TTAAACATGAGTGTTCAAAGACGTCGATATCTGGCGAA |
| Sequence-based reagent | check-HA-PiunH-1 | This paper | PCR primers | TTCGCTTGAGGTACAGCGAA |
| Sequence-based reagent | check-HA-PiunH-2 | This paper | PCR primers | GCTGTTAATGGCAGAGGTAG |
| Software | SAS | SAS | RRID: SCR_008567 | Version 9.2 |

## Animal management

The experimental procedure followed was identical to that previously outlined (*Fu et al., 2024*). Healthy 1-day-old male geese were obtained from Shixing (Guangdong China). These geese were kept in stainless steel cages, with five geese per cage, and were provided with unlimited access to

food and water. The environment was carefully controlled, starting with a temperature of 33 ± 1°C, which was reduced by 2.5 ± 0.5°C weekly until it stabilized at 26°C. Relative humidity levels were maintained between 45% and 60%.

### HUA modeling treatment

The experimental procedure adhered to the methodology described previously (*Fu et al., 2024*). Eighty one-day-old male geese were randomly assigned to two groups: the control (CON) and the HUA groups. From day 1 to 28, the dietary formulations and nutrient levels followed our prior publication (*Fu et al., 2024*). The CON group was provided with a standard diet, while the HUA group received an HCP diet.

### LP and LP with metabolite gavage treatment in an HUA goose model

The experimental procedure adhered to the same methodology as previously described (*Fu et al., 2024*). 120 1-day-old male geese were selected and assigned to two groups: the CON group (A) and the HUA group (B). After 14 days of HUA modeling treatment, the HUA group was further assigned to three subgroups (B, C, and D). From day 14 to 28, A and B received phosphate-buffered saline (PBS), C received LP bacteria resuspended in PBS, and D received LP with metabolites (containing LP at a concentration >1 × 10$^{10}$ CFU). One goose from each replicate in all groups was slaughtered for analysis at 28 days.

### Intraperitoneal injection and gavage in mice

The experimental procedure adhered to the methodology described previously (*Li et al., 2023b*). Thirty-two male Kunming mice (25 ± 2 g), aged 5 weeks, were acquired from the Guangdong Provincial Medical Laboratory Animal Center, Guangdong, China, and housed in an SPF (specific pathogen-free) facility at the Laboratory Animal Center, South China Agricultural University. The mice were kept in stainless steel cages, with each cage housing up to four mice, and provided with sterile food and water ad libitum. The environment was maintained at a constant temperature of 25 ± 1°C with a 12-hr light/dark cycle. A 5-day acclimatization period was observed to allow the mice to adapt to the conditions. All mice received a consistent maintenance diet throughout the study. The 32 mice were divided into four groups based on body weight. The model group (HUA) was administered hypoxanthine (100 mg/kg) orally and potassium oxonate (300 mg/kg) intraperitoneally daily, both dissolved in a 5% sodium carboxymethylcellulose solution. The gavage group received either 39 mg/ml AL (AL group) or 0.2 ml of a fresh bacterial solution (LP group, 1 × 10$^{10}$ CFU/ml) daily, in addition to the modeling treatment. The control group (CON) was given the same dose of sodium carboxymethylcellulose, and both the control and model groups received PBS by gavage. The experiment spanned 19 days, with knee joint thickness measured using vernier calipers on the final day. All mice were humanely euthanized by CO$_2$ overdose followed by cervical dislocation, in accordance with ethical guidelines.

### Sample collection

The experimental procedure followed the same methodology as previously described (*Fu et al., 2024*). Blood samples were collected from each individual, and serum was separated by centrifugation (1200 × *g*, 10 min, 4°C), then stored at −30°C for analysis. The entire heart, liver, kidney, spleen, thymus, pancreas, bursa of Fabricius, muscular stomach, and glandular stomach were collected and weighed. The lengths of the duodenum, jejunum, ileum, and cecum were measured. Liver, kidney, and jejunum (approximately 1–2 cm) were obtained after rinsing with PBS. Cecum and colon chyme were collected and stored at −80°C until further analysis. Animal body weights were monitored regularly throughout the treatment period.

### Analysis of biochemical parameters

The experimental procedure adhered to the same methodology as previously described (*Fu et al., 2024*). Serum UA, XOD, BUN, creatinine, calcium, and phosphorus (P) were measured by biochemical kits, all kits were purchased from Jiancheng (Nanjing, China). IL-1β, TNF-α, and IFN-γ levels were measured by Enzyme-Linked Immunosorbent assay (ELISA) kit (Meimian Industrial, Jiangsu, China) according to the manufacturer's protocol.

## Western blot analysis

Total protein was extracted by RIPA (Sigma-Aldrich) buffer supplemented with protease and phosphatase inhibitors and protein concentration was determined by a BCA Assay Kit (Thermo Fisher Scientific, USA). Equal protein amounts (20 µg) were electrophoresed on 10% sodium dodecyl sulfate–polyacrylamide gel and the separated proteins were transferred onto the polyvinylidene difluoride (PVDF) membranes (Amersham International, GE Healthcare). The membranes were blocked with Tris-buffered saline with Tween (TBST) containing 3% bovine serum albumin (BSA) for 1 hr at room temperature, and then overnight in primary antibody (*Supplementary file 1*) at 4°C. After several times washes in TBST, the membranes were incubated with secondary antibodies in TBST containing 3% BSA for 1 hr at room temperature. Subsequently, the membranes were treated with ECL western blotting substrate (Amersham International, GE Healthcare, Chicago, IL, USA) and imaged using a chemiluminescence detection system (Bio-Rad Laboratories, Hercules, CA, USA). The band intensity was quantified using ImageJ software.

## 16S rRNA sequencing

The experimental procedure followed the same methodology as previously described (*Fu et al., 2024*). Microbial genomic DNA from cecal contents was subjected to 16S rRNA gene sequencing according to the standard protocols of Majorbio (Shanghai, China). The SILVA database was employed to categorize amplicon sequence variants. Alpha diversity was evaluated for each sample to determine bacterial abundance and uniformity. Differences in gut microbial community structure between groups were analyzed using Unifrac-based weighted PCoA, and statistical differences between groups were assessed with ANOSIM analysis. The nonparametric factorial Kruskal–Wallis and rank tests were used to identify bacterial species with statistically significant differences. Subsequently, LDA scores in LEfSe analyses were utilized to estimate the impact of species abundance on these differences. Correlation coefficients between bacterial species and serum metabolomics were calculated using Spearman correlation analysis. Functional pathways predicted by 16S rRNA data were illustrated using PICRUSt software.

## Strains and Media

The experimental procedure followed the same methodology as previously described (*Fu et al., 2024*). The *L. plantarum* strain SQ001 (CGMCC NO 24791) and the standard *E. coli* strain were obtained from the Waterfowl Nutrition Laboratory (South China Agricultural University, Guangdong, China). LB (Luria-Bertani) and MRS (deMan Rogosa Sharpe) media were procured from Huankai (Guangdong, China). Following the collection of cecal chyme, the suspension was cultured across nine gradient dilutions in a clean bench. Single colonies were then picked, cultured in liquid medium, followed by DNA extraction for 16S rRNA sequencing, with strain identification through NCBI comparisons. *E. coli* was cultured in LB liquid medium, while *L. plantarum* was cultured in MRS liquid medium, both incubated at 37°C for 12 hr. A 5% inoculum volume (vol/vol) was introduced and cultured for three consecutive generations. Subsequently, a 10% inoculum volume (vol/vol) was transferred to the medium for further experiments.

## Determination of UA degradation rate

The experimental procedure followed the same methodology as previously described (*Fu et al., 2024*). The UA solution was prepared by dissolving 40 mg of UA (0.4 g/l) in 100 mL of $K_3PO_4$ solution (10 mmol/l, pH 7.0). A standard curve (0.4, 0.2, 0.1, 0.05, and 0.025 g/l; concentration–peak area) was generated using the external standard method. Next, 2 ml of a 20-hr bacterial culture was centrifuged ($n = 3$, $6000 \times g$, 4°C, 10 min). The pellet underwent two washes with 1 ml of sterile PBS before being resuspended in 750 µl of UA solution. This mixture was incubated for 24 hr ($n = 3$, 37°C, 120 r/min).

## Determination of nucleoside degradation rate

The experimental procedure followed the same methodology as previously described (*Fu et al., 2024*). A nucleoside solution was prepared by dissolving 20 mg of inosine (0.2 g/l) and 20 mg of guanosine (0.2 g/l) in 100 ml of $K_3PO_4$ solution (10 mmol/l, pH 7.0). A standard curve (0.2, 0.1, 0.05, 0.025, and 0.0125 g/l; concentration–peak area) was established using the external standard method. Subsequently, 2 ml of a 20-hr bacterial culture was centrifuged ($n = 3$, $6000 \times g$, 4°C, 10 min). The

pellet underwent two washes with 1 ml of sterile PBS before being resuspended in 750 µl of nucleoside solution and incubated for 1, 3, and 6 hr ($n = 3$, 37°C, 120 r/min).

## High-performance liquid chromatography determination

The experimental procedure followed the same methodology as previously described (*Fu et al., 2024*). The solution was centrifuged (4000 × *g*, 4°C, 10 min). Then, 720 µl of the supernatant was mixed with 80 µl of $HClO_4$ (9:1, 0.1 mol/l). After filtering through a membrane (0.22 µm), 20 µl of the mixture was injected into a high-performance liquid chromatography (HPLC) system. The HPLC utilized a C18 reversed-phase column (spursil EP, 250 mm × 4.6 mm, 5 µm). The mobile phase comprised 20 mmol/l potassium dihydrogen phosphate solution (pH 3.0) with 1% methanol. The flow rate was set to 1 ml/min, and the column temperature was maintained at 25°C. The detection wavelength was 254 nm, and the elution time was 30 min.

## Heterologous expression and gene knockout

Oligonucleotide primers were synthesized by DynaScience (Qingdao, China), and sequencing services were provided by Huada (Shenzhen, China). The heterologous expression plasmid p15A-cm-Pgenta-iunH and the gene knockout plasmid p15A-cm-HA-G-iunH-erm-sacB were prepared as previously described (*Fu et al., 2024*). All strains, plasmids, mutants, and primers used in this study are detailed in *Supplementary file 1*.

## Untargeted metabolomics analysis

The experimental procedure adhered to the same methodology as previously described (*Fu et al., 2024*). Metabolites were extracted using an extraction solution composed of methanol, acetonitrile, and water in a 2:2:1 ratio (500 µl total volume), incorporating an isotope-labeled internal standard mixture. The supernatant obtained was used for LC–MS/MS analysis. These analyses were performed with a UHPLC system (Vanquish, Thermo Fisher Scientific) coupled with a UPLC BEH Amide column (2.1 mm × 100 mm, 1.7 µm) and a Q Exactive HFX mass spectrometer (Orbitrap MS, Thermo). The mobile phases used were 25 mmol/l ammonium acetate and 25 mmol/l ammonium hydroxide solution (pH 9.75, A) and acetonitrile (B). The QE HFX mass spectrometer operated in information-dependent acquisition mode, managed by Xcalibur software (Thermo). Raw data were converted to mzXML format using ProteoWizard and analyzed with XCMS for peak detection, extraction, alignment, and integration. Metabolite annotation was subsequently performed using an MS2 database (Biotree, Shanghai, China) with a cutoff annotation threshold of 0.3.

## Statistical analysis

Unless otherwise specified, all data are presented as mean ± standard error of the mean. Statistical analyses were conducted using SAS 9.2 (SAS Inst. Inc, Cary, NC). Comparisons between two groups were made using the unpaired Student's *t*-test or Mann–Whitney *U* test, while one-way ANOVA with Bonferroni's multiple comparisons test was used for comparisons among more than two groups. p values were indicated in the figures as follows: not significant [ns], $p < 0.05$; *, $p < 0.05$; **, $p < 0.01$; ***, $p < 0.001$; ****, $p < 0.0001$.

## Acknowledgements

This study was sponsored by the National Science Fund for Outstanding Young Scholars (32222080), Guangdong Province Natural Science Funds for Distinguished Young Scholar (2022B1515020016), National Science Fund Project of China (32072751), National Key Research Program (2023YFD1301000, 2021YFD1300404, 2022YFD1300400, 2022YFD1301800), Guangdong Basic and Applied Basic Research Foundation (2022B1515130003), Guangdong Chimelong Philanthropic Foundation (CLPF2021007Z), and China Agriculture Research System (CARS-42-15).

## Additional information

#### Competing interests

Xian-Zhi Jiang, Juan Chen: Affiliated with Biotech Co. Ltd; the author has no financial interests to declare. Peng Zhang: Affiliated with Chimelong Group Co; the author has no financial interests to declare. The other authors declare that no competing interests exist.

## Funding

| Funder | Grant reference number | Author |
|---|---|---|
| National Science Fund for Outsanding Young Scholars | 32222080 | Wence Wang |
| Guangdong Province Natural Science Funds for Distinguished Young Scholar | 2022B1515020016 | Wence Wang |
| National Science Fund Project of China | 32072751 | Wence Wang |
| Guangdong Basic and Applied Basic Research Foundation | 2022B1515130003 | Wence Wang |
| Guangdong Chimelong Philanthropic Foundation | CLPF2021007Z | Wence Wang |
| China Agricultural Research System | CARS-42-15 | Wence Wang |
| National Key Research and Development Program | 2023YFD1301000 | Wence Wang |
| National Key Research and Development Program | 2021YFD1300404 | Wence Wang |
| National Key Research and Development Program | 2022YFD1300400 | Wence Wang |
| National Key Research and Development Program | 2022YFD1301800 | Wence Wang |

The funders had no role in study design, data collection, and interpretation, or the decision to submit the work for publication.

## Author contributions

Yang Fu, Data curation, Formal analysis, Investigation, Writing - original draft, Writing - review and editing; Xiao-Dan Luo, Jin-Ze Li, Qian-Yuan Mo, Yue Zhao, Hao-Tong Luo, Dai-Yang Xia, Wei-Qing Ma, Jian-Ying Chen, Li-Hau Wang, Qiu-Yi Deng, Lukuyu Ben, Muhammad Kashif Saleemi, Xian-Zhi Jiang, Juan Chen, Zhen-Ping Lin, Peng Zhang, Hui Ye, Qing-Yun Cao, Yong-Wen Zhu, Lin Yang, Formal analysis, Investigation; Xue Wang, Resources, Formal analysis, Writing - review and editing; You-Ming Zhang, Resources, Formal analysis, Investigation; Kai Miao, Data curation, Formal analysis, Investigation; Qiang Tu, Resources, Formal analysis; Wence Wang, Resources, Funding acquisition, Writing - original draft, Project administration, Writing - review and editing

## Author ORCIDs

Wence Wang (iD) https://orcid.org/0000-0002-1753-7141

## Ethics

All animal procedures were approved by the Animal Experimentation Ethics Committee of the South China Agricultural University (SYXK-2019-0136).

## Decision letter and Author response

Decision letter https://doi.org/10.7554/eLife.100068.sa1
Author response https://doi.org/10.7554/eLife.100068.sa2

## Additional files

### Supplementary files
- Supplementary file 1. Strains, plasmids, mutants, primers, and antibiotics used in this study.
- Reporting standard 1. ARRIVE author checklist.
- MDAR checklist

### Data availability

RNAseq data are available from the NCBI SRA database under the accession number PRJNA1138299 (https://www.ncbi.nlm.nih.gov/sra/PRJNA1138299). Raw data for mice serum metabolomics, L. plantarum metabolomics, and all figures are presented in Dryad (https://doi.org/10.5061/dryad.573n5tbhh).

The following dataset was generated:

| Author(s) | Year | Dataset title | Dataset URL | Database and Identifier |
|---|---|---|---|---|
| Wang W | 2024 | Host-derived Lactobacillus plantarum alleviates hyperuricemia by improving gut microbial community and hydrolase-mediated degradation of purine nucleosides | https://doi.org/10.5061/dryad.573n5tbhh | Dryad Digital Repository, 10.5061/dryad.573n5tbhh |

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
