## [Editor Report]

Multiple studies have shown that the gut microbiota is involved in the metabolism of uric acid and influences systemic uric acid levels. However, the specific bacteria and genes involved in this process are not known. The current study provides convincing evidence that *Lactobacillus* plantarum prevents the synthesis of uric acid through the hydrolysis of nucleosides through the nucleoside hydrolase gene (iunH). This marks a valuable contribution to the field.

---

## [Decision Letter]

**Decision letter after peer review:**

Thank you for submitting your article "Host-derived *Lactobacillus plantarum* alleviates hyperuricemia by improving gut microbial community and hydrolase-mediated degradation of purine nucleosides" for consideration by *eLife*. Your article has been reviewed by 3 peer reviewers, and the evaluation has been overseen by a Reviewing Editor and Wendy Garrett as the Senior Editor.

Essential revisions (for the authors):

1) Check for grammatical errors and consistency.

2) Provide greater methodological details, as requested.

*Reviewer #1 (Recommendations for the authors):*

1. Line 103: Why were goslings selected as modeling animals instead of geese at other growth stages?

2. Line 136: in vitro should be italicised.

3. Line 140: Strain similarity should be expressed as a specific value.

5. Line 152-154: Is this a metabolomics analysis of the culture medium? It suggests a clear formulation.

6. Line 172-173: What is the role of "advantageous metabolites"? What are the metabolites and give examples.

7. Line 198: Why the author did not investigate the other three nucleoside-related genes for further validation? Are there any plans for this?

8. Check for grammatical errors, such as tense issues in Lines 134-135.

9. Ensure P-values are consistently italicized and case-sensitive throughout the text.

*Reviewer #2 (Recommendations for the authors):*

A set of private Recommendations for the Authors that outline how the science and its presentation could be strengthened.

1. Lines 29 and 35: "*Lactobacillus*" and "iunH" should be italicized. Please ensure that accurate gene nomenclature is used consistently.

2. Line 116-120: "P" should be italicized.

3. Line 134 and 287: "suggest" should be "suggested". The results should be in the past tense, please check the tenses throughout the text.

4. Line 185-186: Please verify the sentence "two were identified as 186 nucleoside hydrolases (iunH)."

5. Line 152 and 193: like "6 hours" and "3h". Please standardize the unit format.

6. Supplemental guidance on isolating, culturing, and identifying microorganisms from the cecum to obtain SQ001.

7. In Figure 8, the abundance of *Lactobacillus* plantarum is higher in the LP+ metabolites group compared to the LP+PBS group, in other words, it may have a better regulation of host purine nucleoside metabolism. What is the explanation for this?

8. In Figure 10 e, metabolic LC-MS data showed L-Methionine and L-Targinine in LP treatment are increased. The author should make further explanation in the Discussion section.

*Reviewer #3 (Recommendations for the authors):*

1. Check the bacteria names in the text for accuracy and ensure they are italicized. For example, line 29, 116-119.

2. Please standardize abbreviations consistently, e.g., L. plantarum.

3. "geese" or "gosling", please maintain consistency throughout the text.

4. Check the P-value, e.g., Line 113-120.

5. Check the number of decimal places in the text.

6. Use the past tense when describing the results, e.g., Line 134.

---

## [Author Response]

Essential revisions (for the authors):1) Check for grammatical errors and consistency.

Thank you for your suggestion! We have checked the full article, and corrected in line 125, 144, 164, 177 and 179.

2) Provide greater methodological details, as requested.

We appreciate your careful review of our paper. The Supplementary Material methods for the isolation, culture, and identification of *Lactobacillus* plantarum were added on line 482.

Reviewer #1 (Recommendations for the authors):1. Line 103: Why were goslings selected as modeling animals instead of geese at other growth stages?

We appreciate your careful review of our paper. This model builds on our previous study [1], which indicated that geese are highly susceptible to gout, particularly in young birds, leading to significant morbidity and mortality. In addition, we chose to use young geese to eliminate variability in their initial health status.

[1] Fu, Y., Chen, Y. S., Xia, D. Y., Luo, X. D., Luo, H. T., Pan, J., Ma, W. Q., Li, J. Z., Mo, Q. Y., Tu, Q., Li, M. M., Zhao, Y., Li, Y., Huang, Y. T., Chen, Z. X., Li, Z. J., Bernard, L., Dione, M., Zhang, Y. M., Miao, K., … Wang, W. C. (2024). *Lactobacillus* rhamnosus GG ameliorates hyperuricemia in a novel model. NPJ biofilms and microbiomes, 10(1), 25. https://doi.org/10.1038/s41522-024-00486-9

2. Line 136: in vitro should be italicised.

Thank you! We apologize for this error. We have corrected it and checked the full article to make sure there are no errors.

3. Line 140: Strain similarity should be expressed as a specific value.

Thank you for your close reading of our paper. We added the value of similarity in line 132, which can also be seen in the evolutionary tree (Figure 3a).

5. Line 152-154: Is this a metabolomics analysis of the culture medium? It suggests a clear formulation.

Thank you for your suggestion! This is a metabolomic analysis of the nucleoside solution (inosine, adenosine, and guanosine) following co-culture. We have corrected it in line 144-148.

6. Line 172-173: What is the role of "advantageous metabolites"? What are the metabolites and give examples.

Thank you for your insightful comment. In our study, we found that *Lactobacillus* plantarum, when co-cultured with nucleoside solutions, also produces substances such as linolenic acid and proline, which may also have beneficial effects on the organism in previous studies [1,2]. We have corrected it in line 164-165 and discussed it in line 371.

[1] Fu, Y., Chen, Y. S., Xia, D. Y., Luo, X. D., Luo, H. T., Pan, J., Ma, W. Q., Li, J. Z., Mo, Q. Y., Tu, Q., Li, M. M., Zhao, Y., Li, Y., Huang, Y. T., Chen, Z. X., Li, Z. J., Bernard, L., Dione, M., Zhang, Y. M., Miao, K., … Wang, W. C. (2024). *Lactobacillus* rhamnosus GG ameliorates hyperuricemia in a novel model. NPJ biofilms and microbiomes, 10(1), 25. https://doi.org/10.1038/s41522-024-00486-9

[2] Zhang, MaryAnn et al. “Effect of Dietary and Supplemental Omega-3 Polyunsaturated Fatty Acids on Risk of Recurrent Gout Flares.” Arthritis and rheumatology (Hoboken, N.J.) vol. 71,9 (2019): 1580-1586. doi:10.1002/art.40896

7. Line 198: Why the author did not investigate the other three nucleoside-related genes for further validation? Are there any plans for this?

Thank you for your suggestion. Our in vitro assay showed that nucleosides were hydrolyzed to purine bases, prompting us to focus on the iunH, rihA, and rihC genes. While previous studies have established the nucleoside-hydrolyzing functions of rihA and rihC [1, 2], the function of iunH remains unknown. Thus, we will concentrate on iunH and plan to further investigate the interactions among these three genes.

1. Li M, Wu X, Guo Z, Gao R, Ni Z, Cui H, Zong M, Van Bockstaele F, Lou W. 2023. Lactiplantibacillus plantarum enables blood urate control in mice through degradation of nucleosides in gastrointestinal tract. Microbiome 11:153. 10.1186/s40168-023-01605-y.

2. Li D, Zhang M, Teng ZLA, Lyu Z, Li X, Feng Y, Liu D, Guo Y, Hu Y. 2023. Quercetin-enriched *Lactobacillus* aviarius alleviates hyperuricemia by hydrolase-mediated degradation of purine nucleosides. Pharmacol Res 196:106928. 10.1016/j.phrs.2023.106928.

8. Check for grammatical errors, such as tense issues in Lines 134-135.

Thank you for your close reading of our paper. We rewrote the sentence as “The overall findings of this study suggest that hyperuricemia in geese has a detrimental impact and leads to alterations in the composition of gut microbiota.” in line 125-127.

9. Ensure P-values are consistently italicized and case-sensitive throughout the text.

Thank you for your suggestion! We have corrected it in line 105-110 and checked the full article to make sure there are no errors.

Reviewer #2 (Recommendations for the authors):A set of private Recommendations for the Authors that outline how the science and its presentation could be strengthened.1. Lines 29 and 35: "Lactobacillus" and "iunH" should be italicized. Please ensure that accurate gene nomenclature is used consistently.

Thank you! We apologize for this error. We have corrected it in line 27 and 32, and checked the full article to make sure there are no errors.

2. Line 116-120: "P" should be italicized.

Thank you! We have corrected this error in line 106-110, and checked the full article to make sure there are no errors.

3. Line 134 and 287: "suggest" should be "suggested". The results should be in the past tense, please check the tenses throughout the text.

Thank you for pointing this out. We have corrected this error in line 125 and 279, and checked the full article to make sure there are no errors.

4. Line 185-186: Please verify the sentence "two were identified as 186 nucleoside hydrolases (iunH)."

Thank you! We have verified the sentence "two were identified as 186 nucleoside hydrolases (iunH)", and rewrote the sentence as “Two genes were classified as nucleoside hydrolases (iunH), two as ribonucleoside hydrolases (rihA, rihC), and one as a nucleoside permease (yxjA) (Figure 5c)” in line 179.

5. Line 152 and 193: like "6 hours" and "3h". Please standardize the unit format.

Thank you for pointing this out. We have corrected this error in line 139 and 185 and checked the full article to make sure there are no errors.

6. Supplemental guidance on isolating, culturing, and identifying microorganisms from the cecum to obtain SQ001.

Thank you for your suggestion! We have added the following to our material approach, “Following the collection of cecal chyme, the suspension was cultured across nine gradient dilutions in a clean bench. Single colonies were then picked, cultured in liquid medium, followed by DNA extraction for 16S rRNA sequencing, with strain identification through NCBI comparisons.” In line 485.

7. In Figure 8, the abundance of Lactobacillus plantarum is higher in the LP+ metabolites group compared to the LP+PBS group, in other words, it may have a better regulation of host purine nucleoside metabolism. What is the explanation for this?

Thank you for your close reading of our paper. We concur that metabolites derived from *Lactobacillus* plantarum have the potential to effectively modulate host purine nucleoside metabolism. In response to a previous reviewer's query regarding advantageous metabolites, we have elucidated the presence of beneficial compounds such as proline and linolenic acid in *Lactobacillus* plantarum metabolites, which prior investigations have demonstrated to possess ameliorative properties against hyperuricaemia [1,2].

[1] Fu, Y., Chen, Y. S., Xia, D. Y., Luo, X. D., Luo, H. T., Pan, J., Ma, W. Q., Li, J. Z., Mo, Q. Y., Tu, Q., Li, M. M., Zhao, Y., Li, Y., Huang, Y. T., Chen, Z. X., Li, Z. J., Bernard, L., Dione, M., Zhang, Y. M., Miao, K., … Wang, W. C. (2024). *Lactobacillus* rhamnosus GG ameliorates hyperuricemia in a novel model. NPJ biofilms and microbiomes, 10(1), 25. https://doi.org/10.1038/s41522-024-00486-9

[2] Zhang, MaryAnn et al. “Effect of Dietary and Supplemental Omega-3 Polyunsaturated Fatty Acids on Risk of Recurrent Gout Flares.” Arthritis and rheumatology (Hoboken, N.J.) vol. 71,9 (2019): 1580-1586. doi:10.1002/art.40896

8. In Figure 10 e, metabolic LC-MS data showed L-Methionine and L-Targinine in LP treatment are increased. The author should make further explanation in the Discussion section.

Thank you for your insightful comment. Among the serum metabolites of the animals, we statistically analysed the metabolites with differences, mainly including nucleotide metabolism and amino acid metabolism, and as the reviewer said, the significant up-regulation of L-Methionine and L-Targinine occurred in *Lactobacillus plantarum* gavage group, which might suggest that these two substances mediate the action of *Lactobacillus plantarum*, but unfortunately, in Figure 10, the correlation analysis suggests that it does not have a significant correlation between these two substances and *Lactobacillus plantarum* (Figure 10m), and in Figure 4, the results of the *Lactobacillus plantarum* in vitro assay similarly do not indicate changes in L-Methionine and L-Targinine (Figure 4e and i). We discuss indole-3-carboxylic acid (ICA), which has a significant positive correlation with *Lactobacillus plantarum*, in row 336. The role of metabolites would be an interesting topic for follow-up studies, but this paper focuses more on how *Lactobacillus plantarum* itself is involved in host purine nucleoside metabolism, and therefore there is not much discussion of metabolites.

Reviewer #3 (Recommendations for the authors):1. Check the bacteria names in the text for accuracy and ensure they are italicized. For example, line 29, 116-119.

Thank you! We apologize for this error. We have corrected it in line 27, 106-110, and checked the full article to make sure there are no errors.

2. Please standardize abbreviations consistently, e.g., L. plantarum.

Thank you for your close reading of our paper. We have standardized the abbreviations.

3. "geese" or "gosling", please maintain consistency throughout the text.

Thank you for your suggestion! We've standardized on “goslings” to “geese”.

4. Check the P-value, e.g., Line 113-120.

Thank you! We have corrected this error in line 105-110, and checked the full article to make sure there are no errors.

5. Check the number of decimal places in the text.

Thank you for your suggestion. We have checked the number of decimal places in the text.

6. Use the past tense when describing the results, e.g., Line 134.

Thank you for pointing this out. We have changed “suggest” to “suggested” in line 125.